# Bicriteria Multidimensional Mechanism Design with Side Information

**Maria-Florina Balcan**
School of Computer Science
Carnegie Mellon University
`ninamf@cs.cmu.edu`

**Siddharth Prasad**
Computer Science Department
Carnegie Mellon University
`sprasad2@cs.cmu.edu`

**Tuomas Sandholm**
Computer Science Department
Carnegie Mellon University
Optimized Markets, Inc.
Strategy Robot, Inc.
Strategic Machine, Inc.
`sandholm@cs.cmu.edu`

## Abstract

We develop a versatile new methodology for multidimensional mechanism design that incorporates side information about agent types to generate high social welfare and high revenue simultaneously. Prominent sources of side information in practice include predictions from a machine-learning model trained on historical agent data, advice from domain experts, and even the mechanism designer's own gut instinct. In this paper we adopt a prior-free perspective that makes no assumptions on the correctness, accuracy, or source of the side information. First, we design a meta-mechanism that integrates input side information with an improvement of the classical VCG mechanism. The welfare, revenue, and incentive properties of our meta-mechanism are characterized by novel constructions we introduce based on the notion of a *weakest competitor*, which is an agent that has the smallest impact on welfare. We show that our meta-mechanism, when carefully instantiated, simultaneously achieves strong welfare and revenue guarantees parameterized by errors in the side information. When the side information is highly informative and accurate, our mechanism achieves welfare and revenue competitive with the total social surplus, and its performance decays continuously and gradually as the quality of the side information decreases. Finally, we apply our meta-mechanism to a setting where each agent's type is determined by a constant number of parameters. Specifically, agent types lie on constant-dimensional subspaces (of the potentially high-dimensional ambient type space) that are known to the mechanism designer. We use our meta-mechanism to obtain the first known welfare and revenue guarantees in this setting.

## 1 Introduction

Mechanism design is a high-impact branch of economics and computer science that studies the implementation of socially desirable outcomes among strategic self-interested agents. Major real-world use cases include combinatorial auctions (*e.g.*, strategic sourcing, radio spectrum auctions), matching markets (*e.g.*, housing allocation, ridesharing), project fundraisers, and many more. The two most commonly studied objectives in mechanism design are *welfare maximization* and *revenue maximization*. In many settings, welfare maximization, or *efficiency*, is achieved by the classic

37th Conference on Neural Information Processing Systems (NeurIPS 2023).

Vickrey-Clarke-Groves (VCG) mechanism [19, 28, 50]. Revenue maximization is a much more elusive problem that is only understood in very special cases. The seminal work of Myerson [42] characterized the revenue-optimal mechanism for the sale of a single item in the Bayesian setting, but it is not even known how to optimally sell two items. It is known that welfare and revenue are generally at odds and optimizing one can come at the great expense of the other [1, 4, 7, 23, 33].

In this paper we study how *side information* (or *predictions*) about the agents can help with *bicriteria* optimization of both welfare and revenue. Side information can come from a variety of sources that are abundantly available in practice such as predictions from a machine-learning model trained on historical agent data, advice from domain experts, or even the mechanism designer's own gut instinct. Machine learning approaches that exploit the proliferation of agent data have in particular witnessed a great deal of success both in theory [8, 10, 38, 41] and in practice [24, 25, 46, 52]. In contrast to the typical Bayesian approach to mechanism design that views side information through the lens of a prior distribution over agents, we adopt a prior-free perspective that makes no assumptions on the correctness, accuracy, or source of the side information. A nascent line of work (part of a larger agenda on learning-augmented algorithms [40]) has begun to examine the challenge of exploiting predictions when agents are self-interested, but only for fairly specific problem settings [3, 13–15, 26, 53]. We contribute to this area with a general side-information-dependent meta-mechanism for a wide swath of multidimensional mechanism design problems that aim for high social welfare and high revenue.

## 1.1 Our contributions

Our main contribution is a versatile meta-mechanism that integrates side information about agent types with the bicriteria goal of simultaneously optimizing welfare and revenue.

In Section 2 we formally define the components of multidimensional mechanism design with side information. The abstraction of multidimensional mechanism design is a rich language that allows our theory to apply to many real-world settings including combinatorial auctions, matching markets, project fundraisers, and more—we expand on this list of examples further in Section 2. We also present the weakest-competitor VCG mechanism introduced by Krishna and Perry [34] and prove that it is revenue-optimal among all efficient mechanisms in the prior-free setting (extending their work which was in the Bayesian setting for a fixed known prior).

In Section 3 we present our meta-mechanism for mechanism design with side information. It generalizes the mechanism of Krishna and Perry [34]. We introduce the notion of a weakest-competitor set and a weakest-competitor hull, which are constructions that are crucial to understanding the payments and incentive properties of our meta-mechanism.

In Section 4 we prove that our meta-mechanism—when carefully instantiated—achieves strong welfare and revenue guarantees that are parameterized by errors in the side information. Our mechanism works by independently expanding the input predictions, where the expansion radius for each prediction is drawn randomly from a logarithmic discretization of the diameter of the ambient type space. Our mechanism achieves the efficient welfare OPT and revenue *at least* $\Omega(\text{OPT}/\log H)$ when the side information is highly informative and accurate, where $H$ is an upper bound on any agent's value for any outcome. Its revenue approaches OPT if its initialization parameters are chosen wisely. Its performance decays gradually as the quality of the side information decreases (whereas naïve approaches suffer from huge discontinuous drops in performance). *Prior-free efficient welfare* OPT*, or total social surplus, is the strongest possible benchmark for both welfare and revenue.* Finally, we extend our methods to a more general, more expressive side information language.

In Section 5 we use our meta-mechanism to derive new results in a setting where each agent's type is determined by a constant number of parameters. Specifically, agent types lie on constant-dimensional subspaces (of the potentially high-dimensional ambient type space) that are known to the mechanism designer. *For example, a real-estate agent might infer a buyers' relative property values based on value per square foot.* When each agent's true type is known to lie in a particular $k$-dimensional subspace of the ambient type space, we show how to use our meta-mechanism to guarantee revenue at least $\Omega(\text{OPT}/k(\log H)^k)$ while simultaneously guaranteeing welfare at least $\text{OPT}/\log H$.

Traditionally it is known that welfare and revenue are at odds and maximizing one objective comes at the expense of the other. Our results show that side information can help mitigate this difficulty.

## 1.2 Related work

*Side information in mechanism design.* Various mechanism design settings have been studied under the assumption that some form of public side information is available. Medina and Vassilvitskii [38] study single-item (unlimited supply) single-bidder posted-price auctions with bid predictions. Devanur et al. [22] study the sample complexity of (single-parameter) auctions when the mechanism designer receives a distinguishing signal for each bidder. More generally, the active field of *algorithms with predictions* aims to improve the quality of classical algorithms when machine-learning predictions about the solution are available [40]. There have been recent explicit connections of this paradigm to settings with strategic agents [3, 13–15, 26]. Most related to our work, Xu and Lu [53] study auctions for the sale of a (single copy of a) single item when the mechanism designer receives point predictions on the bidders' values. Unlike our approach, they focus on *deterministic* modifications of a second-price auction. An important drawback of determinism is that revenue guarantees do not decay continuously as prediction quality degrades. For agents with value capped at $H$ there is an error threshold after which, in the worst case, only a $1/H$-fraction of revenue can be guaranteed (this is not even competitive with a vanilla second-price auction). Xu and Lu [53] prove that such a revenue drop is unavoidable by deterministic mechanisms. Finally, our setting is distinct from, but similar to in spirit, work that uses public attributes for market segmentation to improve revenue [8, 11].

*Welfare-revenue tradeoffs in auctions.* Welfare and revenue relationships in Bayesian auctions have been widely studied since the seminal work of Bulow and Klemperer [17]. Welfare-revenue tradeoffs for second-price auctions with reserve prices in the single item setting have been quantified [21, 31], with some approximate understanding of the Pareto frontier [23]. Anshelevich et al. [4] study welfare-revenue tradeoffs in large markets, Aggarwal et al. [2] study the efficiency of revenue-optimal mechanisms, and Abhishek and Hajek [1] study the efficiency loss of revenue-optimal mechanisms.

*Constant-parameter mechanism design.* Revenue-optimal mechanism design for settings where each agent's type space is of a constant dimension has been studied previously in certain specific settings. Single-parameter mechanism design is a well-studied topic dating back to the seminal work of Myerson [42], who (1) characterized the set of all truthful allocation rules and (2) derived the Bayesian optimal auction based on virtual values (a quantity that is highly dependent on knowledge of the agents' value distributions). Archer and Tardos [5] also characterize the set of allocation rules that can be implemented truthfully in the single-parameter setting, and use this to derive polynomial-time mechanisms with strong revenue approximation guarantees in various settings. Kleinberg and Yuan [33] prove revenue guarantees for a variety of single-parameter settings that depend on distributional parameters. Constrained buyers with two-parameter values have also been studied [36, 43].

*Combinatorial auctions for limited supply.* Our mechanism when agent types lie on known linear subspaces can be seen as a generalization of the well-known logarithmic revenue approximation that is achieved by a second-price auction with a random reserve price in the single-item setting [27]. Similar revenue approximations have been derived in multi-item settings for various classes of bidder valuation functions such as unit-demand [30], additive [35, 48], and subadditive [9, 18]. To the best of our knowledge, no previous techniques handle agent types on low-dimensional subspaces. Furthermore, our results are not restricted to combinatorial auctions unlike most previous research.

## 2 Problem formulation, example applications, and weakest-competitor VCG

We consider a general multidimensional mechanism design setting with a finite allocation space $\Gamma$ and $n$ agents. $\Theta_i$ is the type space of agent $i$. Agent $i$'s true private type $\theta_i \in \Theta_i$ determines her value $v(\theta_i, \alpha)$ for allocation $\alpha \in \Gamma$. We will interpret $\Theta_i$ as a subset of $\mathbb{R}^\Gamma$, so $\theta_i[\alpha] = v(\theta_i, \alpha)$. We use $\boldsymbol{\theta} \in \bigtimes_{i=1}^n \Theta_i$ to denote a profile of types and $\boldsymbol{\theta}_{-i} \in \Theta_{-i} := \bigtimes_{j \neq i} \Theta_j$ to denote a profile of types excluding agent $i$. We now introduce our model of side information. For each agent, the mechanism designer receives a subset of the type space predicting that the subset contains the agent's true yet-to-be-revealed type. Formally, the mechanism designer receives additional information about each agent in the form of a refinement of each agent's type space, given by $\widetilde{\Theta}_1 \subseteq \Theta_1, \ldots, \widetilde{\Theta}_n \subseteq \Theta_n$. These refinements postulate that the true type of bidder $i$ is actually contained in $\widetilde{\Theta}_i$ (though the mechanism designer does not necessarily know whether or not these predictions are valid). We refer to the $\widetilde{\Theta}_i$ as *side-information sets* or *predictions*. To simplify exposition, we assume that prior

to receiving side information the mechanism designer has no differentiating information about the agents' types, that is, $\Theta_1 = \cdots = \Theta_n$. Let $\Theta$ denote this common *ambient type space*.

A mechanism with side information is specified by an allocation rule $\alpha(\boldsymbol{\theta}; \widetilde{\Theta}_1, \ldots, \widetilde{\Theta}_n) \in \Gamma$ and a payment rule $p_i(\boldsymbol{\theta}; \widetilde{\Theta}_1, \ldots, \widetilde{\Theta}_n) \in \mathbb{R}$ for each agent $i$. We assume agents have quasilinear utilities. A mechanism is *incentive compatible (IC)* if $\theta_i \in \text{argmax}_{\theta_i' \in \Theta_i} \theta_i[\alpha(\theta_i', \boldsymbol{\theta}_{-i}; \widetilde{\Theta}_1, \ldots, \widetilde{\Theta}_n)] - p_i(\theta_i', \boldsymbol{\theta}_{-i}; \widetilde{\Theta}_1, \ldots, \widetilde{\Theta}_n)$ holds for all $i, \theta_i \in \Theta_i, \boldsymbol{\theta}_{-i} \in \Theta_{-i}, \widetilde{\Theta}_1 \subseteq \Theta_i, \ldots, \widetilde{\Theta}_n \subseteq \Theta_n$, that is, agents are incentivized to report their true type regardless of what other agents report and regardless of the side information used by the mechanism (this definition is equivalent to the usual notion of dominant-strategy IC and simply stipulates that side information ought to be used in an IC manner). A mechanism is *individually rational (IR)* if $\theta_i[\alpha(\theta_i, \boldsymbol{\theta}_{-i}; \widetilde{\Theta}_1, \ldots, \widetilde{\Theta}_n)] - p_i(\theta_i, \boldsymbol{\theta}_{-i}; \widetilde{\Theta}_1, \ldots, \widetilde{\Theta}_n) \geq 0$ holds for all $i, \theta_i, \boldsymbol{\theta}_{-i}, \widetilde{\Theta}_1, \ldots, \widetilde{\Theta}_n$. We will analyze a variety of randomized mechanisms that randomize over IC and IR mechanisms. Such randomized mechanisms are thus IC and IR in the strongest possible sense (as supposed to weaker in-expectation IC/IR). *An important note: no assumptions are made on the veracity of $\widetilde{\Theta}_i$, and agent $i$'s misreporting space is the ambient type space $\Theta_i$.*

### Example applications

Our model of side information within the rich language of multidimensional mechanism design allows us to capture a variety of different problem scenarios where both welfare and revenue are desired objectives. We list a few examples of different multidimensional mechanism settings along with examples of different varieties of side information sets.

- Combinatorial auctions: There are $m$ indivisible items to be allocated among $n$ agents (or to no one). The allocation space $\Gamma$ is the set of $(n+1)^m$ allocations of the items and $\theta_i[\alpha]$ is agent $i$'s value for the bundle of items she is allocated by $\alpha$. Let X and Y denote two of the items for sale. The set $\widetilde{\Theta}_i = \{\theta_i : \theta_i[\{X, Y\}] \geq 9, \theta_i[\{X\}] + \theta_i[\{Y\}] \geq 10\}$ represents the prediction that agent $i$'s values for X and Y individually sum up to at least \$10, and her value for the bundle is at least \$9. Here, $\widetilde{\Theta}_i$ is the intersection of linear constraints.

- Matching markets: There are $m$ items (*e.g.*, houses) to be matched to $n$ buyers. The allocation space $\Gamma$ is the set of matchings on the bipartite graph $K_{m,n}$ and $\theta_i[\alpha]$ is buyer $i$'s value for the item $\alpha$ assigns her. Let $\alpha_1, \alpha_2, \alpha_3$ denote three matchings that match house 1, house 2, and house 3 to agent $i$, respectively. The set $\widetilde{\Theta}_i = \{\theta_i : \theta_i[\alpha_1] = 2 \cdot \theta_i[\alpha_2] = 0.75 \cdot \theta_i[\alpha_3]\}$ represents the information that agent $i$ values house 1 twice as much as house 2, and $3/4$ as much as house 3. Here, $\widetilde{\Theta}_i$ is the linear space given by $\text{span}(\langle 1, 1/2, 4/3 \rangle)$.

- Fundraising for a common amenity: A multi-story office building that houses several companies is opening a new cafeteria on a to-be-determined floor and is raising construction funds. The allocation space $\Gamma$ is the set of floors of the building and $\theta_i[\alpha]$ is the (inverse of the) cost incurred by building-occupant $i$ for traveling to floor $\alpha$. The set $\widetilde{\Theta}_i = \{\theta_i : \|\theta_i - \theta_i^*\|_p \leq k\}$ postulates that $i$'s true type is no more than $k$ away from $\theta_i^*$ in $\ell_p$-distance, which might be derived from an estimate of the range of floors agent $i$ works on based on the company agent $i$ represents. Here, $\widetilde{\Theta}_i$ is given by a (potentially nonlinear) distance constraint.

- Bidding for a shared outcome: A delivery service that offers multiple delivery rates (priced proportionally) needs to decide on a delivery route to serve $n$ customers. The allocation space $\Gamma$ is the set of feasible routes and $\theta_i[\alpha]$ is agent $i$'s value for receiving her packages after the driving delay specified by $\alpha$. Let $\alpha_t$ denote an allocation that imposes a driving delay of $t$ on agent $i$. The set $\widetilde{\Theta}_i = \{\theta_i : \theta_i[\alpha_0] \geq 50, \theta_i[\alpha_{t+1}] \geq f_t(\theta_i[\alpha_t]) \; \forall t\}$ is the prediction that agent $i$ is willing to pay \$50 to receive her package as soon as possible, and is at worst a time discounter determined by (potentially nonlinear) discount functions $f_t$. Here, the complexity of $\widetilde{\Theta}_i$ is determined by the $f_t$.

In our results we will assume that $\Theta = [0, H]^\Gamma$ imposing a cap $H$ on any agent's value for any allocation. This is the only problem-specific parameter in our results. In the above four bulleted examples $H$ represents the maximum value any agent has for the grand bundle of items, any available house, the cafeteria opening on her floor, and receiving her packages with no delay, respectively.

**The weakest-competitor VCG mechanism**

The VCG mechanism can generally be highly suboptimal when it comes to revenue [7, 39, 49] (and conversely mechanisms that shoot for high revenue can be highly welfare suboptimal). However, if efficiency is enforced as a constraint of the mechanism design, then the *weakest-competitor VCG (WCVCG) mechanism* introduced by Krishna and Perry [34] is in fact revenue optimal (they call it the generalized VCG mechanism). While VCG payments are based on participation externalities, WCVCG payments are based on agents being replaced by *weakest competitors* who have the smallest impact on welfare. This approach yields a strict revenue improvement over vanilla VCG. Krishna and Perry [34] proved that the Bayesian version of WCVCG is revenue optimal among all efficient, IC, and IR mechanisms. The (prior-free) WCVCG mechanism works as follows. Given reported types $\boldsymbol{\theta}$, it uses the efficient allocation $\alpha^* = \operatorname{argmax}_{\alpha \in \Gamma} \sum_{i=1}^n \theta_i[\alpha]$. The payments are given by $p_i(\boldsymbol{\theta}) = \min_{\widetilde{\theta}_i \in \Theta_i}(\max_{\alpha \in \Gamma} \sum_{j \neq i} \theta_j[\alpha] + \widetilde{\theta}_i[\alpha]) - \sum_{j \neq i} \theta_j[\alpha^*]$. Here, $\Theta_i$ is the ambient type space of agent $i$. If $0 \in \Theta_i$, $p_i$ is the vanilla VCG payment. Krishna and Perry [34] prove the following result in the Bayesian setting, which we reproduce in a stronger prior-free form for completeness.

**Theorem 2.1.** *Weakest-competitor VCG is revenue-optimal subject to efficiency, IC, and IR.*

Proofs of all results in this paper are in Appendix A. Our meta-mechanism (Section 3) is a generalization of WCVCG that uses side information sets rather than the ambient type space to determine payments. (Misreporting is not limited to side information sets.) Our meta-mechanism relaxes efficiency in order to use the side information to boost revenue.

## 3 Weakest-competitor sets and our meta-mechanism

In this section we present our meta-mechanism for mechanism design with side information. Our meta-mechanism generalizes the WCVCG mechanism. We begin by introducing some new constructions based on the concept of a weakest competitor. These constructions are the key ingredients in understanding the role of side information in our meta-mechanism. Let $\theta \preceq \theta'$ if $\theta[\alpha] \leq \theta'[\alpha]$ for all $\alpha \in \Gamma$. Let $\theta \precsim \theta'$ if $\theta[\alpha] \leq \theta'[\alpha]$ for all $\alpha \in \Gamma$ and there exists $\alpha' \in \Gamma$ with $\theta[\alpha'] < \theta'[\alpha']$. Let $\theta \prec \theta'$ if $\theta[\alpha] < \theta'[\alpha]$ for all $\alpha \in \Gamma$. We assume $\Theta_i = \Theta = [0, H]^\Gamma$ for all $i$, that is, all agents share a common ambient type space with no up-front differentiating information.

**Definition 3.1.** The *extended weakest-competitor set* of a closed set $\widetilde{\Theta}_i$, denoted by $\overline{\mathsf{WC}}(\widetilde{\Theta}_i)$, is the subset of all weakest competitors in $\widetilde{\Theta}_i$ over all possible type profiles of the other agents. Formally, $\overline{\mathsf{WC}}(\widetilde{\Theta}_i) := \{\operatorname{argmin}_{\widetilde{\theta}_i \in \widetilde{\Theta}_i}(\max_{\alpha \in \Gamma} \sum_{j \neq i} \theta_j[\alpha] + \widetilde{\theta}_i[\alpha]) : \boldsymbol{\theta}_{-i} \in \Theta_{-i}\}$. The *weakest-competitor set* of $\widetilde{\Theta}_i$, denoted by $\mathsf{WC}(\widetilde{\Theta}_i)$, is the subset of $\overline{\mathsf{WC}}(\widetilde{\Theta}_i)$ where ties in the argmin are broken by discarding any $\theta'$ in the argmin if there exists $\theta$ also in the argmin with $\theta \precsim \theta'$. We call members of both $\overline{\mathsf{WC}}(\widetilde{\Theta}_i)$ and $\mathsf{WC}(\widetilde{\Theta}_i)$ *weakest competitors* and say $\widehat{\theta}_i$ is a *weakest competitor relative to* $\boldsymbol{\theta}_{-i}$ if $\widehat{\theta}_i \in \operatorname{argmin}_{\widetilde{\theta}_i \in \widetilde{\Theta}_i} \max_{\alpha \in \Gamma} \sum_{j \neq i} \theta_j[\alpha] + \widetilde{\theta}_i[\alpha]$.

The weakest-competitor set is a natural notion of a lower bound corresponding to a given predicted type set. From the perspective of WCVCG, the payment of an agent with true type in $\widetilde{\Theta}_i$ only depends on $\mathsf{WC}(\widetilde{\Theta}_i)$ and not on $\widetilde{\Theta}_i$. Motivated by this observation, we define the weakest-competitor hull, which can be viewed as a "weakest-competitor relaxation".

**Definition 3.2.** The *weakest-competitor hull* of $\widetilde{\Theta}_i$, denoted by $\mathsf{WCH}(\widetilde{\Theta}_i)$, is the maximal set $S$ such that $\mathsf{WC}(S) = \mathsf{WC}(\widetilde{\Theta}_i)$ (no $T \supset S$ satisfies $\mathsf{WC}(T) = \mathsf{WC}(\widetilde{\Theta}_i)$).

Weakest-competitor sets and hulls can be simply characterized without explicit reference to the mechanics of WCVCG. Figure 1 displays examples in a two-dimensional type space.

**Theorem 3.3.** *Let* $\Theta = [0, H]^\Gamma$ *and let* $\widetilde{\Theta} \subseteq \Theta$ *be closed. Then* $\overline{\mathsf{WC}}(\widetilde{\Theta}) = \{\theta \in \widetilde{\Theta} : \{\theta' \in \widetilde{\Theta} : \theta' \prec \theta\} = \emptyset\}$, $\mathsf{WC}(\widetilde{\Theta}) = \{\theta \in \widetilde{\Theta} : \{\theta' \in \widetilde{\Theta} : \theta' \precsim \theta\} = \emptyset\}$, *and* $\mathsf{WCH}(\widetilde{\Theta}) = \{\theta \in \Theta : \exists \theta' \in \widetilde{\Theta} \text{ s.t. } \theta \succeq \theta'\}$ *is the upwards closure of* $\widetilde{\Theta}$.

We now present our meta-mechanism, which we denote by $\mathcal{M}$. It uses the efficient allocation, but that allocation is enjoyed only by the subset of agents able to compete with the weakest competitors in the side information set. $\mathcal{M}$ then implements the weakest-competitor payments on those agents.

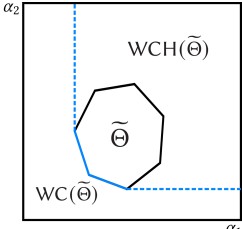 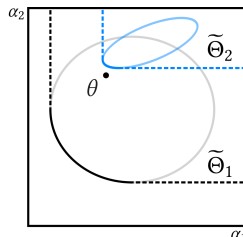

Figure 1: **Left:** Example WC and WCH when $|\Gamma| = 2$. $\overline{\mathsf{WC}}(\widetilde{\Theta})$ is depicted in solid and dashed blue, $\mathsf{WC}(\widetilde{\Theta})$ is depicted in solid blue, and $\mathsf{WCH}(\widetilde{\Theta})$ is the region enclosed by $\overline{\mathsf{WC}}(\widetilde{\Theta})$. **Right:** The prediction $\widetilde{\Theta}_1$ (depicted in gray) is valid ($\gamma^V = 0$), but is highly inaccurate. The prediction $\widetilde{\Theta}_2$ (depicted in light blue) is invalid ($\gamma^V > 0$), but is more accurate than $\widetilde{\Theta}_1$. A small expansion of $\widetilde{\Theta}_2$ would yield a valid and highly accurate prediction.

The input subsets $\widetilde{\Theta}_1, \ldots, \widetilde{\Theta}_n$ represent the side information/predictions given to the mechanism designer that postulate that $\theta_i \in \widetilde{\Theta}_i$.

---

Meta-mechanism $\mathcal{M}$

Input: subsets $\widetilde{\Theta}_1, \ldots, \widetilde{\Theta}_n \subseteq \Theta$ given to mechanism designer.
- Based on $\widetilde{\Theta}_1, \ldots, \widetilde{\Theta}_n$, come up with $\widehat{\Theta}_1, \ldots, \widehat{\Theta}_n$.
Agents asked to reveal types $\theta_1, \ldots, \theta_n$.
- Let $\alpha^* = \operatorname{argmax}_{\alpha \in \Gamma} \sum_{i=1}^n \theta_i[\alpha]$ and for each $i$ let

$$p_i = \min_{\widetilde{\theta}_i \in \mathsf{WC}(\widehat{\Theta}_i)} \left( \max_{\alpha \in \Gamma} \sum_{j \neq i} \theta_j[\alpha] + \widetilde{\theta}_i[\alpha] \right) - \sum_{j \neq i} \theta_j[\alpha^*].$$

- Let $\mathcal{I} = \left\{ i : \theta_i[\alpha^*] - p_i \geq 0 \right\}$. If agent $i \notin \mathcal{I}$, $i$ is excluded and receives zero utility (zero value and zero payment).[a] If agent $i \in \mathcal{I}$, $i$ enjoys allocation $\alpha^*$ and pays $p_i$.

---

[a]One practical consideration is that this step might require a more nuanced implementation of an "outside option" for agents to be indifferent between participating and being excluded versus not participating at all. (We do not pursue this highly application-specific issue in this work.) In auction and matching settings this step is standard and innocuous; the agent simply receives no items.

---

Meta-mechanism $\mathcal{M}$ generates welfare equal to $\sum_{i \in \mathcal{I}} \theta_i[\alpha^*]$ and revenue equal to $\sum_{i \in \mathcal{I}} p_i$. $\mathcal{M}$ does not specify how to set $\widehat{\Theta}_1, \ldots, \widehat{\Theta}_n$ based on $\widetilde{\Theta}_1, \ldots, \widetilde{\Theta}_n$ (hence the "meta" label). This challenge is the subject of the later sections where we will describe, based on the setting, how to set the $\widehat{\Theta}_i$ in order to generate high welfare and high revenue. We now establish the incentive properties of $\mathcal{M}$.

**Theorem 3.4.** $\mathcal{M}$ is IC and IR.

Next we show that WCH precisely captures the set of agent types that never violate IR. This consideration does not arise in WCVCG since in that setting misreporting is limited to the set used in the weakest-competitor minimization, and hence IR is never violated. In our setting, we make no assumptions on the veracity of the sets $\widehat{\Theta}_i$ and must therefore reckon with the possibility that an agent is unable to compete with the weakest competitors in $\mathsf{WC}(\widehat{\Theta}_i)$.

**Theorem 3.5.** Let $\theta_i$ denote the true type of agent $i$ and let $\widehat{\Theta}_1, \ldots, \widehat{\Theta}_n$ denote the side information sets used by $\mathcal{M}$. Then $i \in \mathcal{I}$ for all $\boldsymbol{\theta}_{-i} \iff \theta_i \in \mathsf{WCH}(\widehat{\Theta}_i)$.

Theorem 3.5 shows that $i$ is guaranteed to participate in $\mathcal{M}$ regardless of other agents' types if and only if $\theta_i \in \mathsf{WCH}(\widehat{\Theta}_i)$. We capitalize on this observation when we derive revenue guarantees for $\mathcal{M}$, since the welfare of $\mathcal{M}$ is at least $\sum_{i:\theta_i \in \mathsf{WCH}(\widehat{\Theta}_i)} \theta_i[\alpha^*]$ and its revenue is at least $\sum_{i:\theta_i \in \mathsf{WCH}(\widehat{\Theta}_i)} p_i$.

Before we proceed to our main analyses of the key properties and guarantees of $\mathcal{M}$, we briefly discuss its computational complexity. We consider the special case where the side-information sets are polytopes. Let $size(\widehat{\Theta}_i)$ denote the encoding size of the constraints defining $\widehat{\Theta}_i$.

**Theorem 3.6.** Let $\widehat{\Theta}_i$ be a polytope. Payment $p_i$ in $\mathcal{M}$ can be computed in $poly(|\Gamma|, size(\widehat{\Theta}_i), n)$ time. Furthermore, determining membership in $\mathsf{WCH}(\widehat{\Theta}_i)$ can be done in $poly(|\Gamma|, size(\widehat{\Theta}_i))$ time.

# 4 Main guarantees of the mechanism in terms of prediction quality

In this section we prove our main guarantees on our meta-mechanism $\mathcal{M}$ in terms of the quality of the side information $\widetilde{\Theta}_1, \ldots, \widetilde{\Theta}_n$. We will largely refer to the side information as *predictions* in this section to emphasize that $\widehat{\Theta}_i$ could be wildly incorrect/inaccurate. To state our results we need the following notation which will be used throughout the remainder of the paper. Given agent types $\theta_1, \ldots, \theta_n$, let $\alpha_{\mathsf{opt}}$ denote the efficient allocation among the $n$ agents and let $\mathsf{OPT} = \max_{\alpha \in \Gamma} \sum_{i=1}^n \theta_i[\alpha] = \sum_{i=1}^n \theta_i[\alpha_{\mathsf{opt}}]$ denote the welfare of the efficient allocation (also called the total social surplus). Let $\mathsf{VCG}$ denote VCG revenue on the $n$ agents.

The following lemma shows that payment $p_i$ in $\mathcal{M}$ can be related to agent $i$'s value for $\alpha_{\mathsf{opt}}$ if $i$ has a valid side information set ($\theta_i \in \mathsf{WCH}(\widehat{\Theta}_i)$). We incur a loss term equal to the $\ell_\infty$-Hausdorff distance from the true type $\theta_i$ to $\mathsf{WC}(\widehat{\Theta}_i)$, defined as $d_H(\theta_i, \mathsf{WC}(\widehat{\Theta}_i)) := \max_{\widehat{\theta}_i \in \mathsf{WC}(\widehat{\Theta}_i)} \|\theta_i - \widehat{\theta}_i\|_\infty$.

**Lemma 4.1.** *Run $\mathcal{M}$ with $\widehat{\Theta}_i$. If $i$ is such that $\theta_i \in \mathsf{WCH}(\widehat{\Theta}_i)$, then $p_i \geq \theta_i[\alpha_{\mathsf{opt}}] - d_H(\theta_i, \mathsf{WC}(\widehat{\Theta}_i))$.*

*Measuring the error of a prediction.* Before instantiating $\mathcal{M}$ with specific rules to determine the $\widehat{\Theta}_i$ from the $\widetilde{\Theta}_i$, we define our notions of prediction error, which are motivated by Lemma 4.1.

**Definition 4.2.** The *invalidity* of a prediction $\widetilde{\Theta}_i$, denoted by $\gamma_i^V$, is the distance from the true type $\theta_i$ of agent $i$ to $\mathsf{WCH}(\widetilde{\Theta}_i)$: $\gamma_i^V := d(\theta_i, \mathsf{WCH}(\widetilde{\Theta}_i)) = \min_{\widetilde{\theta}_i \in \mathsf{WCH}(\widetilde{\Theta}_i)} \|\theta_i - \widetilde{\theta}_i\|_\infty$.

**Definition 4.3.** The *inaccuracy* of a prediction $\widetilde{\Theta}_i$ is the quantity $\gamma_i^A := d_H(\theta_i, \mathsf{WC}(\widetilde{\Theta}_i))$.

We say that a prediction $\widetilde{\Theta}_i$ is *valid* if $\gamma_i^V = 0$, that is, $\theta_i \in \mathsf{WCH}(\widetilde{\Theta}_i)$. We say that a prediction is *perfect* if $\gamma_i^A = 0$ or, equivalently, $\mathsf{WC}(\widetilde{\Theta}_i) = \{\theta_i\}$. If a prediction is perfect, then it is also valid. Our main results will depend on these error measures. See Figure 1 for an illustration.

*Consistency and robustness.* We say a mechanism is $(a, b)$-*consistent* and $(c, d)$-*robust* if when predictions are perfect it satisfies $\mathbb{E}[\text{welfare}] \geq a \cdot \mathsf{OPT}$, $\mathbb{E}[\text{revenue}] \geq b \cdot \mathsf{OPT}$, and satisfies $\mathbb{E}[\text{welfare}] \geq c \cdot \mathsf{OPT}$, $\mathbb{E}[\text{revenue}] \geq d \cdot \mathsf{VCG}$ independent of the prediction quality. Consistency demands near-optimal performance when the side information is perfect, and therefore we compete with the total social surplus $\mathsf{OPT}$ on both the welfare and revenue fronts. Robustness deals with the case of arbitrarily bad side information, in which case we would like our mechanism's performance to be competitive with vanilla VCG, which already obtains welfare equal to $\mathsf{OPT}$. High consistency and robustness ratios are in fact trivial to achieve, and we will thus largely not be too concerned with these measures—our main goal is to design high-performance mechanisms that degrade gracefully as the prediction errors increase. In Appendix B we show that the trivial mechanism that discards all side information with probability $\beta$ and trusts the side information completely with probability $1 - \beta$ is $(1, 1 - \beta)$-consistent and $(\beta, \beta)$-robust, but suffers from huge discontinuous drops in performance even when predictions are nearly perfect.

*Random expansion mechanism.* Our guarantees will depend on $H$; an upper bound on any agent's values. This is the only problem-domain-specific parameter in our results (examples are in Section 2). $H = \max_{\theta_1, \theta_2 \in \Theta} \|\theta_1 - \theta_2\|_\infty$ is the $\ell_\infty$-diameter of $\Theta$. For a point $\theta$, let $\mathcal{B}(\theta, r) = \{\theta' : \|\theta - \theta'\|_\infty \leq r\}$ be the closed $\ell_\infty$-ball centered at $\theta$ with radius $r$. For a set $\widetilde{\Theta}$, let $\mathcal{B}(\widetilde{\Theta}, r) = \cup_{\widetilde{\theta} \in \widetilde{\Theta}} \mathcal{B}(\widetilde{\theta}, r)$ denote the $\ell_\infty$-expansion of $\widetilde{\Theta}$ by $r$. For $\zeta_i \geq 0, \lambda_i > 0$, and $K_i := \lceil \log_2((H - \zeta_i)/\lambda_i) \rceil$, let $\mathcal{M}_{\zeta, \lambda}$ denote the mechanism that for each $i$ independently sets

$$\boxed{\widehat{\Theta}_i = \mathcal{B}(\widetilde{\Theta}_i, \zeta_i + 2^{k_i} \cdot \lambda_i), \text{ where } k_i \sim_{\mathsf{unif}} \{0, 1, \ldots, K_i\}.}$$

We now state and discuss our main welfare and revenue guarantees on $M_{\zeta, \lambda}$. Define $\log^+ : \mathbb{R} \to \mathbb{R}_{\geq 0}$ by $\log^+(x) = 0$ if $x \leq 0$ and $\log^+(x) = \max\{0, \log(x)\}$ if $x > 0$.

**Theorem 4.4.** $\mathbb{E}[\text{welfare}] \geq \max\{(1 - \max_i \frac{\lceil \log_2^+((\gamma_i^V - \zeta_i)/\lambda_i) \rceil}{1 + \lceil \log_2((H - \zeta_i)/\lambda_i) \rceil}), \frac{1}{1 + \lceil \max_i \log_2((H - \zeta_i)/\lambda_i) \rceil}\} \mathsf{OPT}$.

**Theorem 4.5** (Revenue bound 1). *Let $\rho_i = 2(\gamma_i^V - \zeta_i)\mathbf{1}(\zeta_i + \lambda_i < \gamma_i^V) + \lambda_i \mathbf{1}(\zeta_i + \lambda_i \geq \gamma_i^V)$. Then* $\mathbb{E}[\text{revenue}] \geq \frac{1}{1 + \lceil \max_i \log_2((H - \zeta_i)/\lambda_i) \rceil}(\mathsf{OPT} - \sum_{i=1}^n (\gamma_i^A + \zeta_i + \rho_i))$.

**Theorem 4.6** (Revenue bound 2). $\mathbb{E}[\text{revenue}] \geq (1 - \max_i \frac{\lceil \log_2^+((\gamma_i^V - \zeta_i)/\lambda_i) \rceil}{1 + \lceil \log_2((H - \zeta_i)/\lambda_i) \rceil})(\mathsf{OPT} - \sum_{i=1}^n (\gamma_i^A + \zeta_i)) - \sum_{i=1}^n \frac{4H}{1 + \lceil \log_2((H - \zeta_i)/\lambda_i) \rceil}$.

*Proof sketch.* $\mathcal{M}_{\zeta,\lambda}$ in essence performs a doubling search with initial hop $\zeta_i$ and $\lambda_i$ controlling how fine-grained the search proceeds. The bounds are proven (Appendix A.3) by controlling participation probabilities $\Pr(\theta_i \in \widehat{\Theta}_i)$ and accounting for conditional payments via Lemma 4.1. $\qquad\square$

First, consider constant $\zeta, \lambda$. Our welfare guarantee degrades from OPT to $\Omega(\text{OPT}/\log H)$ as the invalidity of the predictions increase. Revenue bound 1 degrades from $\Omega(\text{OPT}/\log H)$ as both the invalidity and inaccuracy of the predictions increase. Revenue bound 2 illustrates that we can obtain significantly better performance if the parameters $\zeta, \lambda$ are chosen appropriately. In particular, for any $\varepsilon$, $\zeta_i = \gamma_i^V$ and $\lambda_i \leq O((H - \zeta_i)/2^{H/\varepsilon})$ yields $\mathbb{E}[revenue] \geq \text{OPT} - \sum_{i=1}^{n}(\gamma_i^A + \gamma_i^V + \varepsilon)$, a bound that is only additively worse than the total social surplus (and recovers the total social surplus as $\lambda_i \downarrow 0$ if the predictions are perfect). This bound degrades gradually as $\zeta_i, \lambda_i$ deviate.

To summarize, if the side information is of very high quality, the best parameters $\zeta, \lambda$ nearly recover the total social surplus OPT as welfare *and* revenue, and revenue degrades gradually as the chosen parameters $\zeta, \lambda$ worsen. If the side information is of questionable quality, the best parameters $\zeta, \lambda$ still obtain OPT as welfare, with revenue suffering additively by the prediction errors. As the parameter selection worsens, welfare and revenue degrade to $\Omega(\text{OPT}/\log H)$ with revenue suffering the same additive loss. Effective parameters can be, for example, learned from data [32]. We briefly discuss the consistency and robustness of $\mathcal{M}_{\zeta,\lambda}$ in Appendix B.3, where we show the worst case performance of $\mathcal{M}_{\zeta,\lambda}$ independent of prediction quality is not much worse than vanilla VCG.

## 4.1 More expressive forms of side information

In this subsection we establish two avenues for richer and more expressive side information languages. The first deals with uncertainty and the second with joint multi-agent predictions.

**Uncertainty.** We now show that the techniques we have developed so far readily extend to an even larger more expressive form of side information that allows one to express varying degrees of uncertainty. A *side information structure* corresponding to agent $i$ is given by a partition $(A_1^i, \ldots, A_m^i)$ of the ambient type space $\Theta_i$ into disjoint sets, probabilities $\mu_1^i, \ldots, \mu_m^i \geq 0; \sum_j \mu_j^i = 1$ corresponding to each partition element, and for each partition element an optional probability density function $f_j^i; \int_{A_j^i} f_j^i = 1$. The side information structure represents (1) a belief over what partition element $A_j^i$ the true type $\theta_i$ lies in and (2) if a density is specified, the precise nature of uncertainty over the true type within $A_j^i$. Our model of side information sets $\widetilde{\Theta}_i$ considered earlier in the paper corresponds to the partition $(\widetilde{\Theta}_i, \Theta_i \setminus \widetilde{\Theta}_i)$ with $\mu(\widetilde{\Theta}_i) = 1$ and no specified densities. The richer model allows side information to convey finer-grained beliefs; for example one can express quantiles of certainty, precise distributional beliefs, and arbitrary mixtures of these.

Our notions of prediction error (invalidity and inaccuracy) can be naturally generalized. We define $\overline{\gamma}_i^V = \sum_j \mu_j^i \gamma_i^V(A_j^i; f_j^i)$ and $\overline{\gamma}_i^A = \sum_j \mu_j^i \gamma_i^A(A_j^i; f_j^i)$, where $\gamma_i^V(A_j^i; f_j^i) = d(\theta_i, \text{WCH}(A_j^i))$ if $f_j^i = \text{None}$ and $\gamma_i^V(A_j^i; f_j^i) = \mathbb{E}_{\widetilde{\theta}_i \sim f_j^i}[d(\theta_i, \text{WCH}(\{\widetilde{\theta}_i\}))]$ if $f_j^i$ is a well-defined density. Similarly $\gamma_i^A(A_j^i; f_j^i) = d_H(\theta_i, \text{WC}(A_j^i))$ if $f_j^i = \text{None}$ and $\gamma_i^A(A_j^i; f_j^i) = \mathbb{E}_{\widetilde{\theta}_i \sim f_j^i}[d(\theta_i, \widetilde{\theta}_i)]$ otherwise.

Our generalized version of $M_{\zeta,\lambda}$ first samples a partition element $A_j^i$ according to $(\mu_1^i, \ldots, \mu_m^i)$, and draws $k_i \sim_{\text{unif.}} \{0, \ldots, K_i\}$ where $K_i$ is defined as before. If $f_j^i = \text{None}$, it sets $\widehat{\Theta}_i = \mathcal{B}(A_j^i, \zeta_i + 2^{k_i}\lambda_i)$. Otherwise, it samples $\widetilde{\theta}_i \sim f_j^i$ and sets $\widehat{\Theta}_i = \mathcal{B}(\{\widetilde{\theta}_i\}, \zeta_i + 2^{k_i}\lambda_i)$. Versions of Theorems 4.4, 4.5, and 4.6 carry forward with $\overline{\gamma}_i^V$ and $\overline{\gamma}_i^A$ as the error measures. In Appendix C we provide the derivations, and also take the expressive power one step further by specifying a probability space over $\Theta_i$; its $\sigma$-algebra captures the granularity of knowledge being conveyed and its probability measure captures the uncertainty.

**Joint side information.** So far, side information has been independent across agents. Specifically, the mechanism designer receives sets $\widetilde{\Theta}_i \subseteq \Theta_i$ for each agent $i$ postulating that $\boldsymbol{\theta} = (\theta_1, \ldots, \theta_n) \in \widetilde{\Theta}_1 \times \cdots \times \widetilde{\Theta}_n$. We show that our techniques extend to a more expressive form of side information that allows one to express predictions involving multiple agents. Let $\boldsymbol{\Theta} = \Theta_1 \times \cdots \times \Theta_n$. The mechanism designer receives as side information a set $\widetilde{\boldsymbol{\Theta}} \subseteq \boldsymbol{\Theta}$ postulating that $\boldsymbol{\theta} = (\theta_1, \ldots, \theta_n) \in \widetilde{\boldsymbol{\Theta}}$. Given

an agent $i$, a side information set $\widetilde{\Theta} \subseteq \Theta$, and $\overline{\Theta}_{-i} \subseteq \bigtimes_{j \neq i} \Theta_j$, let $\mathrm{proj}_i(\widetilde{\Theta}; \overline{\Theta}_{-i}) = \{\widetilde{\theta}_i \in \Theta_i :$
$\exists \boldsymbol{\theta}_{-i} \in \overline{\Theta}_{-i} \text{ s.t. } (\widetilde{\theta}_i, \boldsymbol{\theta}_{-i}) \in \widetilde{\Theta}\}$ be the $i$th *projection* of $\widetilde{\Theta}$ with respect to $\overline{\Theta}_{-i}$. The projection
is the set of types for agent $i$ consistent with $\widetilde{\Theta}$ given that the realizations of the other agents' true
types are contained in $\overline{\Theta}_{-i}$. First, if $\widetilde{\Theta}$ is known to be a valid prediction, that is, the true type profile
$\boldsymbol{\theta}$ is guaranteed apriori to lie in $\widetilde{\Theta}$ (equivalently, the joint misreporting space is limited to $\widetilde{\Theta}$), we
generalize the weakest-competitor VCG mechanism as follows. First, agents are asked to reveal their
true types $\boldsymbol{\theta} = (\theta_1, \ldots, \theta_n)$. The allocation used is $\alpha^* = \mathrm{argmax}_{\alpha \in \Gamma} \sum_{i=1}^n \theta_i[\alpha]$ and agent $i$ pays

$$p_i = \min_{\widetilde{\theta}_i \in \mathrm{proj}_i(\widetilde{\Theta}; \theta_{-i})} \left( \max_{\alpha \in \Gamma} \sum_{j \neq i} \theta_j[\alpha] + \widetilde{\theta}_i[\alpha] \right) - \sum_{j \neq i} \theta_j[\alpha^*].$$

This generalized weakest-competitor VCG mechanism is IC, IR, and revenue optimal subject to
efficiency in the joint information setting for the same reason that weakest-competitor VCG is in the
independent information setting.

More generally, given side information set $\widetilde{\Theta}$, our random expansion mechanism can be generalized
as follows. Agents first reveal their true types $\theta_1, \ldots, \theta_n$. For each agent $i$, independently set $\widehat{\Theta}_i = \mathcal{B}(\mathrm{proj}_i(\widetilde{\Theta}, \boldsymbol{\theta}_{-i}), \zeta_i + 2^{k_i}\lambda_i)$. The same guarantees we derived previously hold, with appropriately
modified quality measures: invalidity is $\gamma_i^V = d(\theta_i, \mathsf{WCH}(\mathrm{proj}_i(\widetilde{\Theta}, \boldsymbol{\theta}_{-i})))$ and inaccuracy is $\gamma_i^A = d_H(\theta_i, \mathsf{WC}(\mathrm{proj}_i(\widetilde{\Theta}, \boldsymbol{\theta}_{-i})))$. An important idea highlighted by these mechanisms for joint side
information is that the true types of all agents *other than* $i$ can be heavily utilized in determining
$p_i$. This model of side information affords significantly more expressive power than the agent-
independent model with product structure considered previously. For example, the mechanism
designer might know that sum of the valuations of two customers for a cup of coffee exceeds a
particular threshold, but does not know who has the higher value. Joint side information enables such
a belief to be precisely expressed. It allows the mechanism designer to refine his beliefs on one agent
based on the realized true type of the other agent (which was not possible in our previous framework).

We conclude this section with the observation that these mechanisms can loosely be interpreted
as prior-free quantitative analogues of the seminal total-surplus-extraction Bayesian mechanism
of Crémer and McLean [20] for correlated agents (generalized to infinite type spaces by McAfee and
Reny [37]). This is an interesting connection to explore further in future research.

## 5 Constant-parameter agents: types on low-dimensional subspaces

In this section we show how the theory we have developed so far can be used to derive new revenue
approximation results when the mechanism designer knows that each agent's type belongs to some
low-dimensional subspace of $\mathbb{R}^\Gamma$ (these subspaces can be different for each agent).

This is a slightly different setup from the previous sections. So far, we have assumed that $\Theta_i = \Theta$ for
all $i$, that is, there is an ambient type space that is common to all the agents. Side information sets
$\widetilde{\Theta}_i$ are given as input to the mechanism designer, with no assumptions on quality/correctness (and
our guarantees in Section 4 were parameterized by quality). Here, we assume the side information
that each agent's type lies in a particular subspace is guaranteed to be valid. Two equivalent ways
of stating this setup are (1) that $\Theta_i$ is the corresponding subspace for agent $i$ and the mechanism
designer receives no additional prediction set $\widetilde{\Theta}_i$ or (2) $\Theta_i = \Theta$ for all $i$, $\widetilde{\Theta}_i = \Theta \cap U_i$ where $U_i$ is a
subspace of $\mathbb{R}^\Gamma$, and the mechanism designer has the additional guarantee that $\theta_i \in U_i$ (so $\widetilde{\Theta}_i$ is a
*valid* side-information set). We shall use the language of the second interpretation.

In this setting, while the side information is valid, the inaccuracy errors $\gamma_i^A$ of the sets $\widetilde{\Theta}_i = \Theta \cap U_i$
can be too large to meaningfully use our previous guarantees. In this section we show how to
fruitfully use the information provided by the subspaces $U_1, \ldots, U_n$ within the framework of our
meta-mechanism. We assume $\Theta = [1, H]^\Gamma$, thereby imposing a lower bound of 1 on agent values
(this choice of lower bound is not important, but the knowledge of some lower bound is needed).

Formally, for each $i$, the mechanism designer knows that $\theta_i$ lies in a $k$-dimensional subspace
$U_i = \mathrm{span}(u_{i,1}, \ldots, u_{i,k})$ of $\mathbb{R}^\Gamma$ where each $u_{i,j} \in \mathbb{R}^\Gamma_{\geq 0}$ lies in the non-negative orthant and
$\{u_{i,1}, \ldots, u_{i,k}\}$ is an orthonormal basis for $U_i$. For simplicity, assume $H = 2^a$ for some positive
integer $a$. Let $\mathcal{L}_{i,j} = \{\lambda u_{i,j} : \lambda \geq 0\} \cap [0, H]^\Gamma$ be the line segment that is the portion of the

ray generated by $u_{i,j}$ that lies in $[0, H]^\Gamma$. Let $y_{i,j}$ be the endpoint of $\mathcal{L}_{i,j}$ with $\|y_{i,j}\|_\infty = H$ (the other endpoint of $\mathcal{L}_{i,j}$ is the origin). Let $z_{i,j}^1 = y_{i,j}/2$ be the midpoint of $\mathcal{L}_{i,j}$, and for $\ell = 2, \ldots, \log_2 H$ let $z_{i,j}^\ell = z_{i,j}^{\ell-1}/2$ be the midpoint of $\overline{0z_{i,j}^{\ell-1}}$. So $\|z_{i,j}^{\log_2 H}\|_\infty = 1$. We terminate the halving of $\mathcal{L}_{i,j}$ after $\log_2 H$ steps due to the assumption that $\theta_i \in [1, H]^\Gamma$. For every $k$-tuple $(\ell_1, \ldots, \ell_k) \in \{1, \ldots, \log_2 H\}^k$, let $\widetilde{\theta}_i(\ell_1, \ldots, \ell_k) = \sum_{j=1}^k z_{i,j}^{\ell_j}$. Furthermore, let $W_\ell = \{(\ell_1, \ldots, \ell_k) \in \{1, \ldots, \log_2 H\}^k : \min_j \ell_j = \ell\}$. The sets $W_1, \ldots, W_{\log_2 H}$ partition $\{1, \ldots, \log_2 H\}^k$ into levels, where $W_\ell$ is the set of points with $\ell_\infty$-distance $H/2^\ell$ from the origin. For each agent $i$, our mechanism $\mathcal{M}_k$ independently sets

$$\boxed{\widehat{\Theta}_i = \{\widetilde{\theta}_i(\ell_{i,1}, \ldots, \ell_{i,k})\} \text{ where } \ell_i \sim_{\text{unif.}} \{1, \ldots, \log_2 H\} \text{ and } (\ell_{i,1}, \ldots, \ell_{i,k}) \sim_{\text{unif.}} W_{\ell_i}.}$$

**Theorem 5.1.** $\mathcal{M}_k$ *satisfies* $\mathbb{E}[\textit{welfare}] \geq \mathsf{OPT}/\log_2 H$ *and* $\mathbb{E}[\textit{revenue}] \geq \mathsf{OPT}/(2k(\log_2 H)^k)$.

As mentioned in Section 1, $\mathcal{M}_k$ can be viewed as a generalization of the $\log H$ revenue approximation in the single-item limited-supply setting that is achieved by a second-price auction with a uniformly random reserve price from $\{H/2, H/4, \ldots, 1\}$ [27]. Our results apply not only to auctions but to general multidimensional mechanism design problems such as the examples presented in Section 2.

## 6 Conclusions and future research

We developed a versatile new methodology for multidimensional mechanism design that incorporates side information about agent types with the bicriteria goal of generating high social welfare and high revenue simultaneously. We designed a side-information-dependent meta-mechanism. This mechanism generalizes the weakest-competitor VCG mechanism of Krishna and Perry [34]. Careful instantiations of our meta-mechanism simultaneously achieved strong welfare and revenue guarantees that were parameterized by errors in the side information, and additionally proved to be fruitful in a setting where each agent's type lies on a constant-dimensional subspace (of the potentially high-dimensional ambient type space) that is known to the mechanism designer.

There are many new research directions that stem from our work. First, how far off are our mechanisms from the welfare-versus-revenue Pareto frontier? The weakest-competitor VCG mechanism is one extreme point, but what does the rest of the frontier look like? One possible approach here would be to extend our theory beyond VCG to the larger class of affine maximizers (which are known to contain high-revenue mechanisms)—we provide some initial ideas in Appendix D.

*Computational complexity:* An important facet that we have largely ignored is computational complexity. The computations in our mechanism involving weakest competitors scale with the description complexity of $\widetilde{\Theta}_i$ (*e.g.*, the number of constraints, the complexity of constraints, and so on). An important question here is to understand the computational complexity of our mechanisms as a function of the differing (potentially problem-specific) language structures used to describe the side-information sets $\widetilde{\Theta}_i$. In particular, the classes of side-information sets that are accurate, natural/interpretable, and easy to describe might depend on the specific mechanism design domain. Expressive bidding languages for combinatorial auctions have been extensively studied with massive impact in practice [46, 47]. Can a similar methodology be developed for side information?

*Improved analysis when there is a known prior:* Another direction is to improve the Bayesian revenue analysis of Krishna and Perry when there is a known prior over agents' types. Here, the benchmark would be efficient welfare in expectation over the prior. The (Bayesian) WCVCG mechanism of Krishna and Perry uses weakest competitors with respect to the prior's support to guarantee efficient welfare in expectation, but its revenue could potentially be boosted significantly by compromising on welfare as in our random expansion mechanism. Another direction here is to study the setting when the given prior might be inaccurate. Can our random expansion mechanism be used to derive guarantees that depend on the closeness of the given prior to the true prior? Such questions are thematically related to *robust mechanism design* [16]. Another direction along this vein is to generalize our mechanisms to depend on a known prior over prediction errors.

Finally, the WCVCG mechanism of Krishna and Perry is a strict improvement over the vanilla VCG mechanism, yet it appears to not have been further studied nor applied since its discovery. The weakest-competitor paradigm could have applications in economics and computation more broadly.

## Acknowledgments and Disclosure of Funding

This material is based on work supported by the NSF under grants IIS-1901403, CCF-1733556, CCF-1910321, and SES-1919453, by the ARO under award W911NF2210266, and by Defense Advanced Research Projects Agency under cooperative agreement HR00112020003. S. Prasad thanks Morgan McCarthy for helpful discussions about real-world use cases of multidimensional mechanism design and Misha Khodak for detailed feedback on an earlier draft.

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

# A  Proofs of results

We provide complete proofs of all results from the main body of the paper.

## A.1  Omitted proofs from Section 2

*Proof of Theorem 2.1.* Weakest-competitor VCG is incentive compatible for the same reason that VCG is incentive compatible: the minimization in the payment formula (the *pivot* term) is independent of bidder $i$'s reported type. Concretely, if $\alpha'$ is the welfare-maximizing allocation when bidder $i$ reports $\theta_i'$, bidder $i$'s utility from reporting $\theta_i'$ is $\sum_{j=1}^n \theta_j[\alpha'] - \min_{\widetilde{\theta}_i \in \Theta_i}(\max_\alpha \sum_{j \neq i} \theta_j[\alpha] + \widetilde{\theta}_i[\alpha])$, which is maximized at $\alpha' = \alpha^*$ (which proves incentive compatibility). Furthermore, for each $i$, $\sum_{j=1}^n \theta_j[\alpha^*] - \min_{\widetilde{\theta}_i \in \Theta_i}(\max_\alpha \sum_{j \neq i} \theta_j[\alpha] + \widetilde{\theta}_i[\alpha]) \geq \sum_{j=1}^n \theta_j[\alpha^*] - \max_\alpha \sum_{j=1}^n \theta_j[\alpha] = 0$, which proves individual rationality. The proof that weakest-competitor VCG is revenue optimal follows from the revenue equivalence theorem; the necessary ingredients may be found in the monograph by Vohra [51]. Let $p_i(\boldsymbol{\theta})$ be the weakest-competitor VCG payment rule, and let $p_i'(\boldsymbol{\theta})$ be any other payment rule that also implements the efficient allocation rule. By revenue equivalence, for each $i$, there exists $h_i(\boldsymbol{\theta}_{-i})$ such that $p_i'(\theta_i, \boldsymbol{\theta}_{-i}) = p_i(\theta_i, \boldsymbol{\theta}_{-i}) + h_i(\boldsymbol{\theta}_{-i})$. Suppose $\boldsymbol{\theta}$ is a profile of types such that $p_i'$ generates strictly greater revenue than $p_i$, that is, $\sum_{i=1}^n p_i'(\boldsymbol{\theta}) > \sum_{i=1}^n p_i(\boldsymbol{\theta})$. Equivalently $\sum_{i=1}^n p_i(\theta, \boldsymbol{\theta}_{-i}) + h_i(\boldsymbol{\theta}_{-i}) > \sum_{i=1}^n p_i(\theta_i, \boldsymbol{\theta}_{-i})$. Thus, there exists $i^*$ such that $h_{i^*}(\boldsymbol{\theta}_{-i^*}) > 0$. Now, let

$$\widetilde{\theta}_{i^*} = \underset{\theta_{i^*}' \in \Theta_{i^*}}{\arg\min} \max_{\alpha \in \Gamma} \sum_{j \neq i} \theta_j[\alpha] + \theta_{i^*}'[\alpha]$$

be the *weakest competitor* with respect to $\boldsymbol{\theta}_{-i^*}$. If weakest-competitor VCG is run on the type profile $(\widetilde{\theta}_{i^*}, \boldsymbol{\theta}_{-i^*})$, the agent with type $\widetilde{\theta}_{i^*}$ pays their value for the efficient allocation. In other words, the individual rationality constraint is binding for $\widetilde{\theta}_{i^*}$. Since $h_{i^*}(\boldsymbol{\theta}_{-i^*}) > 0$, $p_i'$ violates individual rationality, which completes the proof. $\square$

## A.2  Omitted proofs from Section 3

*Proof of Theorem 3.3.* Let $i$ denote the index of the agent under consideration with type space $\widetilde{\Theta}$. Let $\theta \in \widetilde{\Theta}$ be a point such that there exists $\theta' \in \widetilde{\Theta}$ with $\theta' \prec \theta$. Then,

$$\max_{\alpha \in \Gamma} \sum_{j \neq i} \theta_j[\alpha] + \theta'[\alpha] < \max_{\alpha \in \Gamma} \sum_{j \neq i} \theta_j[\alpha] + \theta[\alpha]$$

for all $\boldsymbol{\theta}_{-i} \in \Theta_{-i}$. So $\theta \notin \overline{\mathsf{WC}}(\widetilde{\Theta})$, which shows that $\overline{\mathsf{WC}}(\widetilde{\Theta}) \subseteq \{\theta \in \widetilde{\Theta} : \{\theta' \in \widetilde{\Theta} : \theta' \prec \theta\} = \emptyset\}$. To show the reverse containment, let $\theta \in \widetilde{\Theta}$ be such that $\{\theta' \in \widetilde{\Theta} : \theta' \prec \theta\} = \emptyset$. Consider any $\boldsymbol{\theta}_{-i} = (\theta_1, \ldots, \theta_{i-1}, \theta_{i+1}, \ldots, \theta_n)$ such that

$$\sum_{j \neq i} \theta_j[\alpha_1] + \theta[\alpha_1] = \sum_{j \neq i} \theta_j[\alpha_2] + \theta[\alpha_2] = \cdots = \sum_{j \neq i} \theta_j[\alpha_{|\Gamma|}] + \theta[\alpha_{|\Gamma|}].$$

The existence of such a $\boldsymbol{\theta}_{-i}$ can be shown explicitly as follows. Let $j \neq i$ be arbitrary. For all $k \notin \{i, j\}$ set $\theta_k = (0, \ldots, 0)$. Without loss of generality relabel the allocations in $\Gamma$ such that $\theta[\alpha_1] \geq \theta[\alpha_2] \geq \cdots \geq \theta[\alpha_{|\Gamma|}]$. Then, set

$$\theta_j = \big(0, \theta[\alpha_1] - \theta[\alpha_2], \ldots, \theta[\alpha_1] - \theta[\alpha_{|\Gamma|}]\big) \in [0, H]^\Gamma.$$

Then, $\theta$ minimizes

$$\max \left\{ \sum_{j \neq i} \theta_j[\alpha_1] + \theta[\alpha_1], \sum_{j \neq i} \theta_j[\alpha_2] + \theta[\alpha_2], \ldots, \sum_{j \neq i} \theta_j[\alpha_{|\Gamma|}] + \theta[\alpha_{|\Gamma|}] \right\}$$

since any $\theta'$ that attains a strictly smaller value must satisfy $\theta' \prec \theta$ (and no such $\theta'$ exists, by assumption). So $\theta \in \overline{\mathsf{WC}}(\widetilde{\Theta})$, which proves the reverse containment. The characterizations of WC and WCH follow immediately. $\square$

*Proof of Theorem 3.4.* $\mathcal{M}$ is incentive compatible for the exact same reason weakest-competitor VCG is incentive compatible (Theorem 2.1). Individual rationality is an immediate consequence of how $\mathcal{M}$ is defined: all agents with potential individual-rationality violations (those not in $\mathcal{I}$) do not participate and receive zero utility. □

*Proof of Theorem 3.5.* Let $\theta_i$ denote the true type of agent $i$ and let $\boldsymbol{\theta}_{-i} = (\theta_1, \ldots, \theta_{i-1}, \theta_{i+1}, \ldots, \theta_n)$ denote the reported types of the other agents. Suppose $\theta_i \notin \mathsf{WCH}(\widehat{\Theta}_i)$. Then, there exists $\widetilde{\theta}_i \in \overline{\mathsf{WC}}(\widehat{\Theta}_i)$ such that $\widetilde{\theta}_i \succ \theta_i$, and there exists $\boldsymbol{\theta}_{-i}$ such that $\widetilde{\theta}_i \in \operatorname{argmin}_{\widehat{\theta}_i \in \widehat{\Theta}_i} \max_{\alpha \in \Gamma} \sum_{j \neq i} \theta_j[\alpha] + \widehat{\theta}_i[\alpha]$ is a weakest competitor relative to $\boldsymbol{\theta}_{-i}$ (the existence of $\boldsymbol{\theta}_{-i}$ follows from the same reasoning as in the proof of Theorem 3.3). As $\widetilde{\theta}_i \succ \theta_i$, agent $i$'s overall utility will be negative. The utility is unchanged and remains negative if $\widetilde{\theta}_i$ is replaced by $\theta_i^* \in \mathsf{WC}(\widehat{\Theta}_i)$ that is also a weakest competitor relative to $\boldsymbol{\theta}_{-i}$. So we have shown there exists $\boldsymbol{\theta}_{-i}$ such that $i \notin \mathcal{I}$.

Conversely suppose $\theta_i \in \mathsf{WCH}(\widehat{\Theta}_i)$. Then, there exists $\theta_i' \in \mathsf{WC}(\widehat{\Theta}_i)$ such that $\theta_i \succeq \theta_i'$. Let $\boldsymbol{\theta}_{-i}$ be arbitrary. Agent $i$'s utility is $\sum_{j=1}^n \theta_j[\alpha^*] - \min_{\widetilde{\theta}_i \in \mathsf{WC}(\widehat{\Theta}_i)}(\max_\alpha \sum_{j \neq i} \theta_j[\alpha] + \widetilde{\theta}_i[\alpha]) \geq \sum_{j=1}^n \theta_j[\alpha^*] - (\max_\alpha \sum_{j \neq i} \theta_j[\alpha] + \theta_i'[\alpha]) \geq \sum_{j=1}^n \theta_j[\alpha^*] - (\max_\alpha \sum_{j \neq i} \theta_j[\alpha] + \theta_i[\alpha]) = 0$, so $i \in \mathcal{I}$, as desired. □

*Proof of Theorem 3.6.* The weakest competitor in $\widehat{\Theta}_i$ relative to $\boldsymbol{\theta}_{-i}$ is the solution $\widetilde{\theta}_i \in \mathbb{R}^\Gamma$ to the linear program

$$\min \left\{ \gamma \ : \ \begin{array}{l} \widetilde{\theta}_i[\alpha] + \sum_{j \neq i} \theta_j[\alpha] \leq \gamma \ \ \forall \alpha \in \Gamma, \\ \widetilde{\theta}_i \in \widehat{\Theta}_i, \gamma \geq 0 \end{array} \right\}$$

with $|\Gamma| + 1$ variables and $|\Gamma| + size(\widehat{\Theta}_i)$ constraints. Generating the first set of constraints requires computing $\sum_{j \neq i} \theta_j[\alpha]$ for each $\alpha \in \Gamma$, which takes time $\leq n|\Gamma|$.

Checking membership of $\theta_i$ in $\mathsf{WCH}(\widehat{\Theta}_i)$ is equivalent to checking feasibility of a polytope

$$\theta_i \in \mathsf{WCH}(\widehat{\Theta}_i) \iff \left\{ \widetilde{\theta}_i : \widetilde{\theta}_i \in \widehat{\Theta}_i, \theta_i[\alpha] \geq \widetilde{\theta}_i[\alpha] \ \forall \alpha \in \Gamma \right\} \neq \emptyset$$

defined by $size(\widehat{\Theta}_i) + |\Gamma|$ constraints. □

More generally, the complexity of the above two mathematical programs is determined by the complexity of constraints needed to define $\widetilde{\Theta}_i$: for example, if $\widetilde{\Theta}_i$ is a convex set then they are convex programs. Naturally, a major caveat of this brief discussion on computational complexity is that $|\Gamma|$ can be very large (for example, $|\Gamma|$ is exponential in combinatorial auctions).

### A.3  Omitted proofs from Section 4

*Proof of Lemma 4.1.* Let $p_i$ denote the payment collected from agent $i$, where $i$ is an agent such that $\theta_i \in \widehat{\Theta}_i$. Let $\theta_i^*$ be the weakest competitor in $\widehat{\Theta}_i$ with respect to $\boldsymbol{\theta}_{-i}$. The utility for agent $i$ under $\mathcal{M}$ is

$$\begin{aligned}
\theta_i[\alpha_{\mathsf{opt}}] - p_i &= \sum_{j=1}^n \theta_j[\alpha_{\mathsf{opt}}] - \min_{\widehat{\theta}_i \in \widehat{\Theta}_i} \left( \max_{\alpha \in \Gamma} \sum_{j \neq i} \theta_j[\alpha] + \widehat{\theta}_i[\alpha] \right) \\
&= \sum_{j=1}^n \theta_j[\alpha_{\mathsf{opt}}] - \left( \max_{\alpha \in \Gamma} \sum_{j \neq i} \theta_j[\alpha] + \theta_i^*[\alpha] \right) \\
&\leq \sum_{j=1}^n \theta_j[\alpha_{\mathsf{opt}}] - \left( \sum_{j \neq i} \theta_j[\alpha_{\mathsf{opt}}] + \theta_i^*[\alpha_{\mathsf{opt}}] \right) \\
&= \theta_i[\alpha_{\mathsf{opt}}] - \theta_i^*[\alpha_{\mathsf{opt}}] \\
&\leq \max_{\widehat{\theta}_i \in \mathsf{WC}(\widehat{\Theta}_i)} \left\| \theta_i - \widehat{\theta}_i \right\|_\infty = d_H(\theta_i, \mathsf{WC}(\widehat{\Theta}_i)),
\end{aligned}$$

as desired. $\qquad\square$

*Proof of Theorem 4.4.* For each agent $i$, let $k_i^*$ be the smallest $k \in \{0, \ldots, K_i\}$ such that $\theta_i \in$ $\mathsf{WCH}(\mathcal{B}(\widehat{\Theta}_i, \zeta_i + 2^k \cdot \lambda_i))$. Equivalently, $k_i^*$ is the minimal $k$ such that $\gamma_i^V \leq \zeta_i + 2^k \lambda_i$. So $k_i^* = \lceil \log_2^+((\gamma_i^V - \zeta_i)/\lambda_i) \rceil$. We have, using the fact that $\theta_i \in \widehat{\Theta}_i \implies i \in \mathcal{I}$ (Theorem 3.5),

$$\mathbb{E}[\text{welfare}] = \mathbb{E}\left[\sum_{i=1}^n \theta_i[\alpha_{\mathsf{opt}}] \cdot \mathbf{1}(i \in \mathcal{I})\right] \geq \mathbb{E}\left[\sum_{i=1}^n \theta_i[\alpha_{\mathsf{opt}}] \cdot \mathbf{1}(\theta_i \in \widehat{\Theta}_i)\right] = \sum_{i=1}^n \theta_i[\alpha_{\mathsf{opt}}] \cdot \Pr(\theta_i \in \widehat{\Theta}_i)$$

and

$$\Pr(\theta_i \in \widehat{\Theta}_i) = \Pr(k_i \geq k_i^*) = 1 - \Pr(k_i < k_i^*) = 1 - \frac{k_i^*}{1 + K_i} = 1 - \frac{\lceil \log_2^+((\gamma_i^V - \zeta_i)/\lambda_i) \rceil}{1 + \lceil \log_2((H - \zeta_i)/\lambda_i) \rceil}.$$

Therefore

$$\mathbb{E}[\text{welfare}] \geq \left(1 - \max_i \frac{\lceil \log_2^+((\gamma_i^V - \zeta_i)/\lambda_i) \rceil}{1 + \lceil \log_2((H - \zeta_i)/\lambda_i) \rceil}\right) \cdot \mathsf{OPT}.$$

If all predictions are valid, we get $\mathbb{E}[\text{welfare}] = \mathsf{OPT}$. The other term of the bound follows from $\Pr(\theta_i \in \widehat{\Theta}_i) \geq \Pr(k_i = k_i^*) = \frac{1}{1+K_i}$. $\qquad\square$

*Proof of Theorem 4.5.* Let $k_i^*$ be defined as in the proof of Theorem 4.4. We compute expected revenue by computing $\mathbb{E}[p_i]$ for each agent $i$. Let $S = \{i : \theta_i \in \widehat{\Theta}_i\}$ be the (random) set of agents with valid predictions post expansion. We have

$$\mathbb{E}[p_i] \geq \mathbb{E}[p_i \mid k_i = k_i^*] \cdot \Pr(k_i = k_i^*) = \frac{1}{1 + K_i} \cdot \mathbb{E}[p_i \mid k_i = k_i^*].$$

Now $k_i = k_i^* \implies i \in S$, so we may apply the payment bound of Lemma 4.1:

$$\mathbb{E}[p_i \mid k_i = k_i^*] \geq \mathbb{E}\left[\theta_i[\alpha_{\mathsf{opt}}] - d_H(\theta_i, \mathsf{WC}(\mathcal{B}(\widetilde{\Theta}_i, \zeta_i + 2^{k_i^*}\lambda_i))) \mid k_i = k_i^*\right]$$
$$= \theta_i[\alpha_{\mathsf{opt}}] - d_H(\theta_i, \mathsf{WC}(\mathcal{B}(\widetilde{\Theta}_i, \zeta_i + 2^{k_i^*}\lambda_i))).$$

Next, we bound $d_H(\theta_i, \mathsf{WC}(\mathcal{B}(\widetilde{\Theta}_i, \zeta_i + 2^{k_i^*}\lambda_i)))$. Let $\widetilde{\theta}_i \in \mathsf{WC}(\mathcal{B}(\widetilde{\Theta}_i, \zeta_i + 2^{k_i^*}\lambda_i))$ be arbitrary. By Lemma A.1 (the statement and proof are at the end of Appendix A.3), $\widetilde{\theta}_i \in \mathsf{WC}(\mathcal{B}(\mathsf{WC}(\widetilde{\Theta}_i), \zeta_i + 2^{k_i^*}\lambda_i))$, so there exists $\theta_i' \in \mathsf{WC}(\widetilde{\Theta}_i)$ such that $\|\widetilde{\theta}_i - \theta_i'\|_\infty \leq \zeta_i + 2^{k_i^*}\lambda$. Moreover, $\|\theta_i - \theta_i'\|_\infty \leq \gamma_i^A$ by definition of $\gamma_i^A$. The triangle inequality therefore yields

$$\left\|\theta_i - \widetilde{\theta}_i\right\|_\infty \leq \left\|\widetilde{\theta}_i - \theta_i'\right\|_\infty + \left\|\theta_i - \theta_i'\right\|_\infty \leq \gamma_i^A + \zeta_i + 2^{k_i^*}\lambda_i,$$

so, as $\widetilde{\theta}_i \in \mathsf{WC}(\mathcal{B}(\widetilde{\Theta}_i, \zeta_i + 2^{k_i^*}\lambda_i))$ was arbitrary, $d_H(\theta_i, \mathsf{WC}(\mathcal{B}(\widetilde{\Theta}_i, r_i^*))) \leq \gamma_i^A + \zeta_i + 2^{k_i^*}\lambda_i$. Now, we claim $2^{k_i^*}\lambda_i \leq \rho_i$. To show this, consider two cases. If $\zeta_i + \lambda_i \geq \gamma_i^V$, that is, $k_i^* = 0$, then $2^{k_i^*}\lambda_i = \lambda_i = \rho_i$. If $\zeta_i + \lambda_i < \gamma_i^V$, then $k_i^* > 0$ and we have $\zeta_i + 2^{k_i^*-1}\lambda_i < \gamma_i^V \leq \zeta + 2^{k_i^*}\lambda_i$, so $2^{k_i^*}\lambda < 2(\gamma_i^V - \zeta_i) = \rho_i$. So

$$d_H(\theta_i, \mathsf{WC}(\mathcal{B}(\widetilde{\Theta}_i, \zeta_i + 2^{k_i^*}\lambda_i))) \leq \gamma_i^A + \zeta_i + \rho_i.$$

Finally, we have

$$\mathbb{E}[p_i] \geq \frac{1}{1 + K_i}\left(\theta_i[\alpha_{\mathsf{opt}}] - (\gamma_i^A + \zeta_i + \rho_i)\right),$$

so

$$\mathbb{E}[\text{revenue}] = \mathbb{E}\left[\sum_{i=1}^n p_i\right] = \sum_{i=1}^n \mathbb{E}[p_i] \geq \sum_{i=1}^n \frac{1}{1 + K_i}\left(\theta_i[\alpha_{\mathsf{opt}}] - (\gamma_i^A + \zeta_i + \rho_i)\right)$$

$$\geq \frac{1}{1 + \lceil \max_i \log_2(\frac{H - \zeta_i}{\lambda_i}) \rceil}\left(\mathsf{OPT} - \sum_{i=1}^n (\gamma_i^A + \zeta_i + \rho_i)\right),$$

as desired. $\qquad\square$

*Proof of Theorem 4.6.* Let $k_i^*$ be defined as in Theorems 4.4 and 4.5. We bound $\mathbb{E}[p_i]$ similarly to the approach in the proof of Theorem 4.5, but account for all possible values of $k_i$ (rather than only conditioning on $k_i = k_i^*$). If $k_i < k_i^*$, then agent $i$ does not participate and pays nothing. We have

$$\mathbb{E}[p_i] = \sum_{k=k_i^*}^{K_i} \mathbb{E}[p_i|k_i = k] \cdot \Pr(k_i = k) \geq \frac{1}{1+K_i} \sum_{k=k_i^*}^{K_i} \left( \theta_i[\alpha_{\mathsf{opt}}] - d_H(\theta_i, \mathsf{WC}(\mathcal{B}(\widetilde{\Theta}_i, \zeta_i + 2^k \cdot \lambda_i))) \right)$$

$$\geq \frac{1}{1+K_i} \sum_{k=k_i^*}^{K_i} \left( \theta_i[\alpha_{\mathsf{opt}}] - (\gamma_i^A + \zeta_i + 2^k \cdot \lambda_i) \right)$$

$$= \left( 1 - \frac{k_i^*}{1+K_i} \right) \left( \theta_i[\alpha_{\mathsf{opt}}] - \gamma_i^A - \zeta_i \right) - \frac{\lambda_i}{1+K_i} \sum_{k=k_i^*}^{K_i} 2^k$$

$$\geq \left( 1 - \frac{k_i^*}{1+K_i} \right) \left( \theta_i[\alpha_{\mathsf{opt}}] - \gamma_i^A - \zeta_i \right) - \frac{\lambda 2^{K_i+1}}{1+K_i}$$

where in the second inequality we have used the the bound $d_H(\theta_i, \mathsf{WC}(\mathcal{B}(\widetilde{\Theta}_i, \zeta_i + 2^k \lambda_i))) \leq \gamma_i^A + \zeta_i + 2^k \lambda_i$, which was derived in the proof of Theorem 4.5. We have $2^{K_i+1} \leq \lambda_i (4(H - \zeta_i)/\lambda_i) \leq 4H$. Substituting and summing over agents yields the desired revenue bound. $\quad\square$

**Lemma A.1.** $\mathsf{WC}(\mathcal{B}(\widetilde{\Theta}, r)) = \mathsf{WC}(\mathcal{B}(\mathsf{WC}(\widetilde{\Theta}), r))$.

*Proof.* We first prove the forwards containment. For the sake of contradiction suppose there exists $\widetilde{\theta} \in \mathsf{WC}(\mathcal{B}(\widetilde{\Theta}, r))$ such that $\widetilde{\theta} \notin \mathsf{WC}(\mathcal{B}(\mathsf{WC}(\widetilde{\Theta}), r))$. Then, there exists $\theta' \in \mathcal{B}(\mathsf{WC}(\widetilde{\Theta}), r)$ such that $\theta' \preccurlyeq \widetilde{\theta}$. But

$$\mathsf{WC}(\widetilde{\Theta}) \subseteq \widetilde{\Theta} \implies \mathcal{B}(\mathsf{WC}(\widetilde{\Theta}), r) \subseteq \mathcal{B}(\widetilde{\Theta}, r),$$

so $\theta' \in \mathcal{B}(\widetilde{\Theta}, r)$ and $\theta' \preccurlyeq \widetilde{\theta}$ which contradicts the assumption that $\widetilde{\theta} \in \mathsf{WC}(\mathcal{B}(\widetilde{\Theta}, r))$.

We now prove the reverse containment. For the sake of contradiction suppose there exists $\widetilde{\theta} \in \mathsf{WC}(\mathcal{B}(\mathsf{WC}(\widetilde{\Theta}), r))$ such that $\widetilde{\theta} \notin \mathsf{WC}(\mathcal{B}(\widetilde{\Theta}, r))$. Then, there exists $\theta' \in \mathcal{B}(\widetilde{\Theta}, r)$ such that $\theta' \preccurlyeq \widetilde{\theta}$. Furthermore, if $\theta' \notin \mathsf{WC}(\mathcal{B}(\widetilde{\Theta}, r))$, there exists $\theta'' \in \mathsf{WC}(\mathcal{B}(\widetilde{\Theta}, r))$ such that $\theta'' \preccurlyeq \theta' \preccurlyeq \theta'$ (if $\theta' \in \mathsf{WC}(\mathcal{B}(\widetilde{\Theta}, r))$, set $\theta'' = \theta'$). From the forward inclusion, we have

$$\mathsf{WC}(\mathcal{B}(\widetilde{\Theta}, r)) \subseteq \mathsf{WC}(\mathcal{B}(\mathsf{WC}(\widetilde{\Theta}), r)) \subseteq \mathcal{B}(\mathsf{WC}(\widetilde{\Theta}), r),$$

so $\theta'' \in \mathcal{B}(\mathsf{WC}(\widetilde{\Theta}), r)$ and $\theta'' \preccurlyeq \widetilde{\theta}$ which contradicts the assumption that $\widetilde{\theta} \in \mathsf{WC}(\mathcal{B}(\mathsf{WC}(\widetilde{\Theta}), r))$.
$\quad\square$

### A.4 Omitted proofs from Section 5

To prove Theorem 5.1 we need the following more direct bound on $p_i$ in terms of the weakest competitor's value.

**Lemma A.2.** *Run $\mathcal{M}$ with $\widehat{\Theta}_i$. If $i$ is such that $\theta_i \in \mathsf{WCH}(\widehat{\Theta}_i)$, then $p_i \geq \widetilde{\theta}_i[\alpha_{\mathsf{opt}}]$ where $\widetilde{\theta}_i$ is the weakest competitor in $\widehat{\Theta}_i$ relative to $\boldsymbol{\theta}_{-i}$.*

*Proof of Lemma A.2.* Truncating the proof of Lemma 4.1 yields the desired statement. $\quad\square$

*Proof of Theorem 5.1.* We have $\mathbb{E}[\text{welfare}] \geq \sum_{i=1}^n \theta_i[\alpha_{\mathsf{opt}}] \cdot \Pr(\theta_i \in \widehat{\Theta}_i) \geq \sum_{i=1}^n \theta_i[\alpha_{\mathsf{opt}}] \cdot \Pr(\ell_i = \log_2 H) = \frac{1}{\log_2 H} \cdot \mathsf{OPT}$ (since $\theta_i \succeq \widetilde{\theta}_i(\log_2 H, \ldots, \log_2 H)$). The proof of the revenue guarantee relies on the following key claim: for each agent $i$, there exists $\ell_{i,1}^*, \ldots, \ell_{i,k}^* \in \{1, \ldots, \log_2 H\}$ such that $\widetilde{\theta}(\ell_{i,1}^*, \ldots, \ell_{i,k}^*) \succeq \frac{1}{2}\theta_i$. To show this, let $\theta_i^j$ denote the projection of $\theta_i$ onto $u_j$, so $\theta_i = \sum_{j=1}^k \theta_i^j$ since $\{u_{i,1}, \ldots, u_{i,k}\}$ is an orthonormal basis. Let $\ell_{i,j}^* = \min\{\ell : \theta_i^j \succeq z_{i,j}^\ell\}$. Then, $z_{i,j}^{\ell_{i,j}^*} \succeq \frac{1}{2}\theta_i^j$, so

$$\widetilde{\theta}(\ell_{i,1}^*, \ldots, \ell_{i,k}^*) = \sum_{j=1}^k z_{i,j}^{\ell_{i,j}^*} \succeq \sum_{j=1}^k \frac{1}{2}\theta_i^j = \frac{1}{2}\theta_i.$$

We now bound the expected payment of agent $i$ as in the previous results. Let $\ell_i^* = \min_j \ell_{i,j}^*$. We have

$$\mathbb{E}[p_i] \geq \mathbb{E}\big[p_i \mid (\ell_{i,1}, \ldots, \ell_{i,k}) = (\ell_{i,1}^*, \ldots, \ell_{i,k}^*)\big] \cdot \Pr\left((\ell_{i,1}, \ldots, \ell_{i,k}) = (\ell_{i,1}^*, \ldots, \ell_{i,k}^*)\right)$$

$$= \frac{1}{|W_{\ell_i^*}| \log_2 H} \cdot \mathbb{E}\big[p_i \mid (\ell_{i,1}, \ldots, \ell_{i,k}) = (\ell_{i,1}^*, \ldots, \ell_{i,k}^*)\big]$$

$$\geq \frac{1}{\log_2 H((\log_2 H)^k - (\log_2 H - 1)^k)} \cdot \mathbb{E}\big[p_i \mid (\ell_{i,1}, \ldots, \ell_{i,k}) = (\ell_{i,1}^*, \ldots, \ell_{i,k}^*)\big]$$

$$\geq \frac{1}{k(\log_2 H)^k} \cdot \mathbb{E}\big[p_i \mid (\ell_{i,1}, \ldots, \ell_{i,k}) = (\ell_{i,1}^*, \ldots, \ell_{i,k}^*)\big]$$

since the probability of obtaining the correct weakest competitor $\widetilde{\theta}(\ell_{i,1}^*, \ldots, \ell_{i,k}^*)$ can be written as the probability of drawing the correct "level" $\ell_i^* \in \{1, \ldots, \log_2 H\}$ times the probability of drawing the correct weakest competitor within the correct level $W_{\ell_i^*}$. We bound the conditional expectation with Lemma A.2,

$$\mathbb{E}\big[p_i \mid (\ell_{i,1}, \ldots, \ell_{i,k}) = (\ell_{i,1}^*, \ldots, \ell_{i,k}^*)\big] \geq \widetilde{\theta}_i(\ell_{i,1}^*, \ldots, \ell_{i,k}^*)[\alpha_{\mathsf{opt}}] \geq \frac{1}{2} \cdot \theta_i[\alpha_{\mathsf{opt}}].$$

Finally,

$$\mathbb{E}[\text{revenue}] = \sum_{i=1}^n \mathbb{E}[p_i] \geq \frac{1}{2k(\log_2 H)^k} \cdot \sum_{i=1}^n \theta_i[\alpha_{\mathsf{opt}}] = \frac{1}{2k(\log_2 H)^k} \cdot \mathsf{OPT},$$

as desired. $\qquad\square$

# B Consistency and robustness

In this section we show that it is trivial to obtain high consistency and robustness ratios with an otherwise undesirable mechanism that yields poor revenue even if predictions are nearly perfect. We also discuss the consistency and robustness of our main mechanism $\mathcal{M}_{\zeta,\lambda}$.

For $S \subseteq \{1, \ldots, n\}$, let $\mathsf{OPT}_S = \sum_{i \in S} \theta_i[\alpha_{\mathsf{opt}}]$ be the welfare generated by the efficient allocation restricted to agents in $S$. Let $\mathsf{VCG}(S)$ denote the revenue of the vanilla VCG mechanism when run among the agents in $S$. Let $\mathsf{VCG}(\beta) = \mathbb{E}[\mathsf{VCG}(S)]$ where $S \subseteq \{1, \ldots, n\}$ is sampled by including each agent in $S$ independently with probability $\beta$. In general, $S \subseteq T \implies \mathsf{VCG}(S) \leq \mathsf{VCG}(T)$ [44], so $\mathsf{VCG}(\beta)$ need not be increasing in $\beta$, but there are various sufficient conditions when revenue monotonicity does hold [6, 29, 54].

## B.1 First approach: trust predictions completely

The first basic instantiation of our meta mechanism $\mathcal{M}$ is the following: the mechanism designer simply sets $\widehat{\Theta}_i = \mathsf{WCH}(\widetilde{\Theta}_i)$ for all $i$. Let $V = \{i : \theta_i \in \mathsf{WCH}(\widetilde{\Theta}_i)\} = \{i : \gamma_i^V = 0\}$ denote the set of agents with valid predictions. The welfare of this mechanism is simply $\mathsf{OPT}_V$ and its revenue is bounded by Lemma 4.1. If all predictions are valid and perfect, that is, $\mathsf{WC}(\widetilde{\Theta}_i) = \{\theta_i\}$ for all $i$, both welfare and revenue are equal to $\mathsf{OPT}$. However, if all predictions are such that $\theta_i \notin \mathsf{WCH}(\widetilde{\Theta}_i)$, both welfare and revenue potentially drop to 0. So this mechanism is $(1, 1)$-consistent and $(0, 0)$-robust.

## B.2 Second approach: discard predictions randomly

The issue with the above mechanism is that if all predictions are invalid, it generates no welfare and no revenue. We show how randomization can quell that issue. One trivial solution is to discard all predictions with probability $\beta$, and trust all predictions completely with probability $(1 - \beta)$. That is, with probability $\beta$ set $(\widehat{\Theta}_1, \ldots, \widehat{\Theta}_n) = (\Theta, \ldots, \Theta)$ and with probability $1 - \beta$ set $(\widehat{\Theta}_1, \ldots, \widehat{\Theta}_n) = (\mathsf{WCH}(\widetilde{\Theta}_1), \ldots, \mathsf{WCH}(\widetilde{\Theta}_n))$, and then run $\mathcal{M}$. This mechanism achieves strong consistency and robustness ratios. Let $V = \{i : \theta_i \in \mathsf{WCH}(\widetilde{\Theta}_i)\}$ be the set of valid predictions. From Lemma 4.1, we have $\mathbb{E}[\text{welfare}] = \beta \cdot \mathsf{OPT} + (1 - \beta) \cdot \mathsf{OPT}_V$ and $\mathbb{E}[\text{revenue}] \geq$

$\beta \cdot \mathsf{VCG}(1) + (1 - \beta) \cdot \left( \mathsf{OPT}_V - \sum_{i \in V} \gamma_i^A \right)$, and thus obtain $(1, 1 - \beta)$-consistency and $(\beta, \beta)$-robustness.

This approach suffers from a major issue: its revenue drops drastically the moment predictions are invalid ($\gamma_i^V > 0$). In particular, if predictions are highly accurate but very slightly invalid (such as the blue prediction in Figure 1), this approach completely misses out on any payments from such agents and drops to the revenue of VCG (which can be drastically smaller than OPT). But, a tiny expansion of these predictions would have sufficed to increase revenue significantly and perform competitively with OPT. One simple approach is to set $\widehat{\Theta}_i$ to be an expansion of $\widetilde{\Theta}_i$ by a parameter $\eta_i$ with some probability, and discard the prediction with complementary probability. If $\gamma_i^V \leq \eta_i$ for all $i$, then such a mechanism would perform well. The main issue with such an approach is that the moment $\gamma_i^V > \eta_i$, our expansion by $\eta_i$ fails to capture the true type $\theta_i$ and the performance drastically drops. Our main mechanism $\mathcal{M}_{\zeta,\lambda}$ essentially selects the $\eta_i$ randomly from a suitable discretization of the ambient type space to be able to capture $\theta_i$ with reasonable probability.

### B.3 Consistency and robustness of $\mathcal{M}_{\zeta,\lambda}$

In this discussion we assume constant $\zeta, \lambda > 0$. Assuming revenue monotonicity, since $\theta_i \in \widehat{\Theta}_i$ with probability at least $\Omega(1/\log H)$, the revenue of our mechanism is never worse than $\mathsf{VCG}(\Omega(1/\log H))$. Thus, in the language of algorithms-with-predictions, $\mathcal{M}_{\zeta,\lambda}$ is $(1, \Omega(1/\log H))$-consistent and, assuming revenue monotonicity, $(\Omega(1/\log H), \mathsf{VCG}(\Omega(1/\log H)))/\mathsf{VCG}(1))$-robust. If VCG revenue is submodular, the robustness ratio is $\geq \Omega(1/\log H)$ (but in general revenue can shrink by more than this ratio [12]). In contrast to the trivial approach that either trusted the side information completely or discarded predictions completely, our random expansion approach does not suffer from large discontinuous drops in welfare nor revenue.

Furthermore, the previous approaches had no way of gracefully dealing with invalid prediction sets. In particular, if $\widetilde{\Theta}_i$ is an invalid prediction, even if a tiny expansion of $\widetilde{\Theta}_i$ would have captured $\theta_i$ (such as the blue set in Figure 1), we gave up on getting any meaningful revenue from agent $i$. When all predictions were invalid (that is, $\gamma_i^V > 0$ for all $i$), our guarantee dropped to $\beta \cdot \mathsf{VCG}(1)$. The random expansion of predictions remedies these issues. Its revenue is nearly $\log H$-competitive with OPT as long as $\sum_i \gamma_i^A + 2\gamma_i^V$ is not too large. In particular, if we have high-accuracy but invalid predictions that are just a small expansion away from capturing $\theta_i$, the mechanism in this section is nearly $\log H$-competitive with OPT whereas the mechanism from the previous section is compared to vanilla VCG due to invalid predictions.

## C An expressive language for side information

A *side information structure* is a probability space $(\Theta, \mathcal{F}, \mu)$ where the ambient type space $\Theta$ is the sample space, $\mathcal{F}$ is a $\sigma$-algebra on $\Theta$, and $\mu$ is a probability measure. (We suppress the agent index for brevity.)

$\mathcal{F}$ induces a partition of $\Theta$ into equivalence classes where $\theta \equiv \theta'$ if $\mathbf{1}(\theta \in A) = \mathbf{1}(\theta' \in A)$ for all $A \in \mathcal{F}$ (so the side-information structure does not distinguish between $\theta$ and $\theta'$). Let $A_\theta = \{\theta' : \theta \equiv \theta'\} \in \mathcal{F}$ be the equivalence class of $\theta$. In this way the $\sigma$-algebra determines the granularity of knowledge being conveyed by the side information structure, and the probability measure $\mu : \mathcal{F} \to [0, 1]$ establishes uncertainty over this knowledge.

We define invalidity and inaccuracy of a side information structure in the natural way. Define random variables $X_i^V, X_i^A : \Theta \to \mathbb{R}_{\geq 0}$ by

$$X_i^V(\theta) = \gamma_i^V(A_\theta) = d(\theta_i, \mathsf{WCH}(A_\theta))$$

and

$$X_i^A(\theta) = \gamma_i^A(A_\theta) = d_H(\theta_i, \mathsf{WC}(A_\theta)).$$

$X_i^V$ and $X_i^A$ are $\mathcal{F}$-measurable since they are (by definition) constant on all atoms of $\mathcal{F}$ (sets $A \in \mathcal{F}$ such that no nonempty $B \subsetneq A$ is in $\mathcal{F}$). The invalidity/inaccuracy distributions on $\mathbb{R}_{\geq 0}$ are given by

$$\Pr(a \leq X_i^V \leq b) = \mu(\{\theta \in \Theta : a \leq \gamma_i^V(A_\theta) \leq b\}) = \mu(\cup\{A_\theta : a \leq \gamma_i^V(A_\theta) \leq b\}).$$

The generalized version of $M_{\zeta,\lambda}$ that receives as input a side information structure for each agent $i$ given by $(\Theta_i, \mathcal{F}_i, \mu_i)$ works as follows. It samples $\widetilde{\theta}_i \sim \Theta_i$ according to $(\mathcal{F}_i, \mu_i)$ and draws $k_i \sim_{\text{unif.}} \{0, \ldots, K_i\}$ where $K_i$ is defined as before. It then sets

$$\widehat{\Theta}_i = \mathcal{B}\left(A_{\widetilde{\theta}_i}, \zeta_i + 2^{k_i}\lambda_i\right).$$

Executing the same analysis in the proofs of Theorems 4.4, 4.5, and 4.6 for a fixed $\widetilde{\theta}_i$, then taking expectation over the draw of $\widetilde{\theta}_i$ yields similar guarantees with $\gamma_i^V$ and $\gamma_i^A$ replaced by $\mathbb{E}[X_i^V]$ and $\mathbb{E}[X_i^A]$, respectively. To show this, we loosen the bounds in Theorems 4.4, 4.5, and 4.6 slightly to make the multiplicative terms convex. As $X_i^V \geq 0$, we have

$$\left\lceil \log_2^+ \left(\frac{X_i^V - \zeta_i}{\lambda_i}\right)\right\rceil = \left\lceil \max\left(0, \log_2\left(\frac{X_i^V - \zeta_i}{\lambda_i}\right)\right)\right\rceil \leq 1 + \log_2\left(1 + \frac{X_i^V - \zeta_i}{\lambda_i}\right)$$

and therefore

$$\mathbb{E}\left[1 - \frac{\lceil \log_2^+((X_i^V - \zeta_i)/\lambda_i)\rceil}{1 + \lceil \log_2((H - \zeta_i)/\lambda_i)\rceil}\right] \geq \mathbb{E}\left[1 - \frac{1 + \log_2(1 + (X_i^V - \zeta_i)/\lambda_i)}{1 + \lceil \log_2((H - \zeta_i)/\lambda_i)\rceil}\right]$$

$$\geq 1 - \frac{1 + \log_2(1 + (\mathbb{E}[X_i^V] - \zeta_i)/\lambda_i)}{1 + \lceil \log_2((H - \zeta_i)/\lambda_i)\rceil}$$

by Jensen's inequality. The corresponding versions of Theorems 4.4, 4.5, and 4.6 follow.

**Theorem C.1.** $\mathbb{E}[\textit{welfare}] \geq \max\{(1 - \max_i \frac{1 + \log_2(1 + (\mathbb{E}[X_i^V] - \zeta_i)/\lambda_i)}{1 + \lceil \log_2((H - \zeta_i)/\lambda_i)\rceil}), \frac{1}{1 + \lceil \max_i \log_2((H - \zeta_i)/\lambda_i)\rceil}\}\mathsf{OPT}.$

**Theorem C.2** (Revenue bound 1). *Let* $\rho_i = \mathbb{E}[2(X_i^V - \zeta_i)\mathbf{1}(\zeta_i + \lambda_i < X_i^V) + \lambda_i \mathbf{1}(\zeta_i + \lambda_i \geq X_i^V)].$ *Then* $\mathbb{E}[\textit{revenue}] \geq \frac{1}{1 + \lceil \max_i \log_2((H - \zeta_i)/\lambda_i)\rceil}(\mathsf{OPT} - \sum_{i=1}^n(\mathbb{E}[X_i^A] + \zeta_i + \mathbb{E}[\rho_i])).$

**Theorem C.3** (Revenue bound 2). $\mathbb{E}[\textit{revenue}] \geq (1 - \max_i \frac{1 + \log_2(1 + (\mathbb{E}[X_i^V] - \zeta_i)/\lambda_i)}{1 + \lceil \log_2((H - \zeta_i)/\lambda_i)\rceil})(\mathsf{OPT} - \sum_{i=1}^n(\gamma_i^A + \zeta_i)) - \sum_{i=1}^n \frac{4H}{1 + \lceil \log_2((H - \zeta_i)/\lambda_i)\rceil}.$

## D  Beyond the VCG mechanism: affine maximizers

Given agent-specific multipliers $\omega = (\omega_1, \ldots, \omega_n) \in \mathbb{R}_{\geq 0}$ and an allocation-based boost function $\lambda : \Gamma \to \mathbb{R}_{\geq 0}$, we define the following meta-mechanism $\mathcal{M}(\omega, \lambda)$ which is a generalization of our meta-mechanism $\mathcal{M}$. The mechanism designer receives as input $\widetilde{\Theta}_1, \ldots, \widetilde{\Theta}_n$, and based on these decides on prediction sets $\widehat{\Theta}_1, \ldots, \widehat{\Theta}_n$. The agents are then asked to reveal their true types $\theta_1, \ldots, \theta_n$. The allocation used is

$$\alpha_{\omega,\lambda} = \operatorname*{argmax}_{\alpha \in \Gamma} \sum_{i=1}^n \omega_i \theta_i[\alpha] + \lambda(\alpha).$$

Let

$$p_i = \frac{1}{\omega_i}\left[\min_{\widetilde{\theta}_i \in \widehat{\Theta}_i}\left(\max_{\alpha \in \Gamma} \sum_{j \neq i} \omega_j \theta_j[\alpha] + \omega_i \widetilde{\theta}_i[\alpha] + \lambda(\alpha)\right) - \left(\sum_{j \neq i} \omega_j \theta_j[\alpha_{\omega,\lambda}] + \lambda(\alpha_{\omega,\lambda})\right)\right].$$

Let

$$\mathcal{I} = \{i : \theta_i[\alpha_{\omega,\lambda}] - p_i \geq 0\}.$$

Agents in $i$ enjoy allocation $\alpha_{\omega,\lambda}$ and pay $p_i$. Agents not in $i$ do not participate and receive zero utility.

This mechanism is the natural generalization of the affine-maximizer mechanism [45] parameterized by $\omega, \lambda$ to our setting. The special case where agent misreporting is limited to $\widehat{\Theta}_i$ is the natural generalization of the weakest-competitor VCG mechanism of Krishna and Perry [34] to affine-maximizer mechanisms. The following is a simple consequence of the proofs that $\mathcal{M}$ and the affine-maximizer mechanism parameterized by $\omega, \lambda$ are incentive compatible and individually rational.

**Theorem D.1.** *For any* $\omega \in \mathbb{R}_{\geq 0}^n$ *and* $\lambda : \Gamma \to \mathbb{R}_{\geq 0}$, $\mathcal{M}(\omega, \lambda)$ *is incentive compatible and individually rational.*

Let $\mathsf{OPT}(\omega, \lambda) = \sum_{i=1}^{n} \theta_i[\alpha_{\omega,\lambda}]$ be the welfare of the $(\omega, \lambda)$-efficient allocation. All of the guarantees satisfied by $\mathcal{M}$ carry over to $\mathcal{M}(\omega, \lambda)$, the only difference being the modified benchmark of $\mathsf{OPT}(\omega, \lambda)$. Of course, $\mathsf{OPT}(\omega, \lambda) \leq \mathsf{OPT}$ is a weaker benchmark than the welfare of the efficient allocation. However, the class of affine maximizer mechanisms is known to achieve much higher revenue than the vanilla VCG mechanism. We leave it as a compelling open question to derive even stronger guarantees on mechanisms of the form $\mathcal{M}(\omega, \lambda)$ when the underlying affine maximizer is known to achieve greater revenue than vanilla VCG.

