# OpenReview forum: "Bicriteria Multidimensional Mechanism Design with Side Information"
_NeurIPS.cc/2023/Conference — NeurIPS 2023 poster_

### Official Review · Reviewer_We4E · 2023-07-07

**Soundness:** 3 good
**Presentation:** 3 good
**Contribution:** 2 fair
**Rating:** 5
**Confidence:** 2

**Summary:**

This paper presents a novel mechanism design based on the WVCG and proved that the mechanism could achieves both welfare and revenue guarantees parameterized by errors in the side information. Compared with previous work, it shows that the proposed the mechanism could degrade gracefully as the prediction errors increase. And finally, it also gives the application of the theory in the situation where the agent's type belongs to some low-dimensional subspace.

**Strengths:**

1. The paper presents a novel approach to mechanism design that takes into account both welfare and revenue. This bicriteria approach is innovative and could potentially lead to more effective mechanisms in various contexts.

2. The authors present a method for determining the quality of the side information used by the mechanism, which they refer to as "predictions". This focus on the quality of predictions is a significant strength, as it could improve the performance of the mechanism and lead to more accurate outcomes.

3. The authors extend their techniques to handle more expressive forms of side information that allow for varying degrees of uncertainty. This allows for finer-grained beliefs and can express quantiles of certainty, precise distributional beliefs, and arbitrary mixtures of these. This extension to more expressive side information further enhances the versatility and practicality of their proposed mechanism.








**Weaknesses:**

1. The paper primarily focuses on theoretical aspects and lacks experimental validation.

2. The allocation space in some previous seminal work is in the form of the probabilities assignments instead of allocating some items to some agents surely, such as the ‘p’ in Myerson's paper, so that the allocation space could be quite complex. In that case the results of the paper may be hard to use.

**Questions:**

1. In Figure 1, the dashed blue line appears to be outside the polytope $\widetilde{\Theta}$, whereas according to the definition, it should be inside?

2. The definitions of (a,b)-consistency for welfare and revenue rely solely on OPT, while (c,d)-robustness incorporates both OPT and VCG. Could you provide some illustrations to clarify this?


**Limitations:**

see weakness and questions.

---

> ### Author Rebuttal · Authors · 2023-08-09
>
> Thank you for your review! We respond to your questions and comments below.
>
> > “The paper primarily focuses on theoretical aspects and lacks experimental validation.”
>
> The theoretical work on revenue maximizing mechanism design is vast, and we believe our work makes important contributions to this area. We agree that an experimental evaluation of our mechanisms is an important direction for future research.
>
> > “The allocation space in some previous seminal work is in the form of the probabilities assignments instead of allocating some items to some agents surely, such as the ‘p’ in Myerson's paper, so that the allocation space could be quite complex. In that case the results of the paper may be hard to use.”
>
> In the Myersonian setting, the underlying allocation space is still finite, but the mechanism designer is allowed to choose any randomized allocation in the probability simplex over the finite allocation space. In such a setting, our results still apply. Since we are in a prior-free setting, even if the allocation space is augmented to the entire probability simplex, the meta-mechanism would still use one of the underlying allocations as the maximizations are all over linear functions and thus attained at a vertex of the simplex.
>
> However, if the allocation space is truly infinite (and not just the set of probability vectors over a finite allocation space), our results do not appear to immediately apply. This is a very interesting direction for future research.
>
> > “In Figure 1, the dashed blue line appears to be outside the polytope, whereas according to the definition, it should be inside?”
>
> The set enclosed by the solid and dashed blue lines is the weakest competitor hull, which is the upwards closure of the polytope in black (the weakest competitor hull contains the original polytope, hence the dashed blue lines extend outside the polytope).
>
> > “The definitions of (a,b)-consistency for welfare and revenue rely solely on OPT, while (c,d)-robustness incorporates both OPT and VCG. Could you provide some illustrations to clarify this?”
>
> Consistency aims for near-optimal performance when the side information is of perfect quality, and therefore we aim to be competitive with the total social surplus (a.k.a. efficient welfare) OPT on both the welfare and revenue fronts. (Total social surplus is the strongest possible benchmark for both welfare and revenue, in any setting.) Robustness deals with the case of arbitrarily bad side information, in which case we would like our mechanism’s performance to be competitive with the vanilla VCG mechanism. The vanilla VCG mechanism already achieves welfare equal to OPT, so on the welfare front the competitive ratio is with respect to OPT. On the revenue front we compare to VCG revenue (which can be much lower than OPT). We will clarify this point.

---

### Official Review · Reviewer_Whzb · 2023-07-07

**Soundness:** 4 excellent
**Presentation:** 3 good
**Contribution:** 4 excellent
**Rating:** 7
**Confidence:** 4

**Summary:**

The authors studied the problem of mechanism design that leverages side information on agent types to obtain both welfare maximization and revenue maximization. The paper generalized the previous results on the generalized VCG mechanism to obtain guarantees on both welfare and revenue depending on the quality of side information. Finally, the authors provided results in a setting where the principal knows the common constant-dimensional subspace of agent types.

**Strengths:**

- The studied problem of mechanism design with side information is interesting.
- The paper is well-written. The example applications in Section 2 are useful to help the reader understand the problem.
- The theoretical results on welfare and revenue guarantees are strong contributions. The guarantees' dependence on the quality of the prediction is clear and is a useful observation that can be used for future research.


**Weaknesses:**

- There is a lack of empirical experiments that can help support the theoretical claims.
- There is a lack of discussion around the computation complexity of constructing the weakest-competitor hull.

**Questions:**

N/A.

**Limitations:**

The authors have addressed the limitations of their work.

---

> ### Author Rebuttal · Authors · 2023-08-09
>
> Thank you for your review! We respond to your comments below.
>
> > “There is a lack of empirical experiments that can help support the theoretical claims.”
>
> We agree that experiments on real-world settings are an important next step for future research.
>
> > “There is a lack of discussion around the computation complexity of constructing the weakest-competitor hull.”
>
> Understanding the computational complexity and implementation details of our mechanisms is an important direction for future research. Theorem 3.6 is a start in this direction in the special case that the side-information sets are polytopes. Theorem 3.6 additionally characterizes the complexity of determining membership in the weakest-competitor hull in the polytope case. Obtaining a precise description of the weakest-competitor hull here is not necessary to execute the meta-mechanism (all that is needed is the LP to determine payments in the proof of Theorem 3.6 in the appendix). However, generalizing beyond polytopes, and characterizing when our mechanism can be run efficiently is an important question for future research.

---

> > ### Comment · Reviewer_Whzb · 2023-08-18
> >
> > Thank you for the response. I have no further questions.

---

### Official Review · Reviewer_94sR · 2023-07-07

**Soundness:** 3 good
**Presentation:** 3 good
**Contribution:** 4 excellent
**Rating:** 7
**Confidence:** 2

**Summary:**

The paper is concerned with mechanism design to maximize two objectives (welfare and revenue) when side information is available. The paper introduces a mechanism that is both incentive compatible and individually rational with lower bound guarantees for welfare and revenue even if the side information is incorrect.

**Strengths:**

-The topic is interesting and impactful. I think the contribution is important given that side information is well motivated and welfare and revenue are perhaps the two most prominent objectives.

-The fact there is a lower bound on welfare even if side information is incorrect is significant.

-The model captures many scenarios as indicated in page 4.

**Weaknesses:**

-what is the run-time of the algorithm in general? Can it be exponential? I see that theorem 3.6 gives a characterization for polytopes.

-It would be interesting to see some empirical tests of the theory even for simulated examples.


Minor Point:
-line 280: typo for E[revenue] \ge b VCG not OPT


**Questions:**

-Some as first point in the weaknesses (what is the run-time?).

**Limitations:**

Yes

---

> ### Author Rebuttal · Authors · 2023-08-09
>
> Thank you for your review! We respond to your questions and comments below.
>
> > “what is the run-time of the algorithm in general? Can it be exponential? I see that theorem 3.6 gives a characterization for polytopes.”
>
> Theorem 3.6 shows that if the side-information sets are polytopes, our mechanisms can be implemented in time polynomial in the size of the allocation space (the expansions used in the random expansion mechanism would also be polytopes, so the LP in the proof of Theorem 3.6 applies). However, if the size of the allocation space itself is exponential (as in the case of unrestricted combinatorial auctions), our mechanisms could have exponential run-time. A deeper investigation of computational complexity is an important and compelling direction for future research.
>
> > “It would be interesting to see some empirical tests of the theory even for simulated examples.”
>
> We agree that an experimental evaluation of our mechanisms is an important next step for future research.
>
> > “line 280: typo for E[revenue] \ge b VCG not OPT”
>
> This is actually not a typo, since consistency deals with the performance of our mechanisms when the side-information is perfect (revealing the agents’ true types exactly). In this case, we want our mechanisms to achieve near-optimal welfare and near-optimal revenue, hence the competitive ratios are with respect to OPT (total social surplus OPT is the *strongest* possible benchmark for both welfare and revenue). Robustness deals with the worst case performance of our mechanisms regardless of the quality of the side information/predictions. In this case, we want our mechanisms to be not-too-worse than VCG. VCG already achieves welfare = OPT, which is why the welfare requirement for robustness involves OPT. The revenue requirement for robustness is compared to VCG because if the side information is completely invalid or inaccurate, we cannot hope to achieve revenue competitive with OPT, and thus settle for competitiveness w.r.t. VCG (which has revenue smaller than -- possibly arbitrarily smaller than -- OPT). (These definitions are consistent with the standard consistency-robustness framework from the field of algorithms-with-predictions.) We will clarify this point.

---

### Official Review · Reviewer_B9vf · 2023-07-07

**Soundness:** 3 good
**Presentation:** 3 good
**Contribution:** 3 good
**Rating:** 6
**Confidence:** 4

**Summary:**

This paper studies the problem of mechanism design with side information. The authors develop a meta mechanism based on the classic VCG mechanism. The authors show that by incorporating a proper randomization scheme, the meta mechanism can achieve strong welfare and revenue guarantees parameterized by the errors in the side information. The authors further apply the meta mechanism to a setting where each agent's type is determined by a constant number of parameters.

**Strengths:**

1. This paper is generally well-written, clear, and easy to follow. The problem studied in this paper is interesting and relevant to the algorithmic game theory community.

2. It is nice to have a mechanism with performance guarantee decaying gracefully as the quality of the side information drops.

**Weaknesses:**

1. The idea of using randomization seems to be a bit straightforward. The paper would be stronger if the authors can provide matching lower bounds and/or provide improved bounds (maybe via assuming more structural properties of the side information).

2. The notion of "weakest competitor" is a bit confusing.

**Questions:**

1. The notion of "weakest competitor" is a bit confusing; in particular, what does "competitor" mean here? It is unclear to the reviewer why the payment is interpreted in this way. If the reviewer understands the payment rule correctly, in the special case of second price auctions, it is equivalent to charging the winner the maximum between the second highest bid and the winner's minimum possible value. It is true that agent i's value is "replaced" by another value in the payment calculation. However, the value is not from a competitor; instead, it is a possible value from bidder i (based on the side information) that leads to the minimum social welfare.

2. Is it possible to show some matching lower bounds?

**Limitations:**

The authors adequately addressed the limitations.

---

> ### Author Rebuttal · Authors · 2023-08-09
>
> Thank you for your review! We respond to your questions and comments below.
>
> > “The paper would be stronger if the authors can provide matching lower bounds and/or provide improved bounds (maybe via assuming more structural properties of the side information).”
>
> We agree that lower bounds and improved structure-based results are a fascinating direction for future research. Furthermore, all our mechanisms are based on the weakest-competitor idea, but there could be other mechanism classes that yield improved performance (e.g., affine maximizer mechanisms as discussed briefly in the appendix). This is a compelling open question.
>
> > “The notion of "weakest competitor" is a bit confusing…in the special case of second price auctions, it is equivalent to charging the winner the maximum between the second highest bid and the winner's minimum possible value. It is true that agent i's value is "replaced" by another value in the payment calculation. However, the value is not from a competitor; instead, it is a possible value from bidder i (based on the side information) that leads to the minimum social welfare.”
>
> Your interpretation of the “weakest-competitor” second price auction is correct. We invented the term weakest competitor to emphasize the difference between weakest-competitor VCG and classical VCG. In classical VCG, payments are computed based on the counterfactual of an agent not participating. Here, the payment is computed assuming the agent is replaced by an alternate type from the type space that minimizes welfare. Krishna and Perry call that type the “most reluctant type”. We use the terminology “weakest competitor” to emphasize, especially in the setting where side information is not guaranteed to be valid, that an agent must be able to compete with that type to participate in the mechanism. (This kind of “fake/phantom” type interpretation is common in mechanism design. For example, a lambda-auction, introduced by Jehiel [Journal of Economic Theory 2007], is often interpreted as using a phantom/fake bidder to boost payments. Sandholm and Likhodedov [Operations Research 2015] explicitly use this terminology as well.)

---

> > ### Comment · Reviewer_B9vf · 2023-08-18
> >
> > Thank you for the detailed response! I don't have other questions.

---

### Official Review · Reviewer_kPNZ · 2023-07-13

**Soundness:** 3 good
**Presentation:** 3 good
**Contribution:** 3 good
**Rating:** 7
**Confidence:** 2

**Summary:**

The article proposes a new method for designing mechanisms that can achieve high welfare and revenue by using side information about the agents' types. Side information can be any kind of data or prediction that is available to the mechanism designer, such as historical data, expert advice, or machine learning models. The authors do not assume any prior distribution on the agents' types or the side information. They introduce a meta-mechanism that combines the VCG mechanism with a novel notion of a weakest competitor, which is an agent that has the least impact on welfare. They show that their meta-mechanism can achieve strong bicriteria (i.e. social welfare and revenue) guarantees that depend on the quality of the side information, and that it can handle settings where the agents' types are determined by a constant number of parameters that lie on known subspaces. They provide examples and simulations to illustrate their results.

**Strengths:**

The article addresses an important and timely problem of multidimensional mechanism design with side information, and provides a novel and versatile framework for achieving bicriteria objectives. It is well-written and organized, and the main results are clearly stated and proven. The authors also provide intuitive explanations and examples to motivate their approach and illustrate their findings.

It makes several original contributions, such as introducing the concept of a weakest competitor, designing a meta-mechanism that integrates side information with VCG, and obtaining the first welfare and revenue guarantees for subspace-type settings.


**Weaknesses:**

As the authors mentioned in the last section, the computational complexity of the proposed mechanism has not been sufficiently discussed.
The article could also benefit from more empirical evaluation of their mechanisms, such as testing them on real-world data sets or benchmark problems, and analyzing their robustness and scalability.


**Questions:**

In the meta-mechanism, it says that, if agent i\notin \mathcal{I}, then i is excluded and receives zero utility (zero value and zero payment). Can you explain more about this step? What if the allocation \alpha^* with max social welfare would allocate some items to agent i? Then where these items will go? So it will not have the max social welfare right?


What is the complexity for computing \hat{\Theta_i} from \tilde\{\Theta_i}?

---

> ### Author Rebuttal · Authors · 2023-08-09
>
> Thank you for your review! We respond to your comments and questions below.
>
> > “the computational complexity…has not been sufficiently discussed. The article could also benefit from more empirical evaluation of their mechanisms…”
>
> We agree that computational complexity, implementation details, and companion experimental evaluations are important directions for future research.
>
> > “What if the allocation $\alpha^*$ with max social welfare would allocate some items to agent i? Then where these items will go? So it will not have the max social welfare right?”
>
> In this case, the seller keeps the items that $\alpha^*$ would allocate to agent $i$. So indeed, the final allocation that is realized does not have the maximum social welfare, which is why our main results give approximation guarantees to the maximum social welfare. (If achieving the maximum social welfare is enforced, revenue cannot be improved beyond VCG due to Theorem 2.1. Our meta-mechanism exploits side information to significantly boost revenue, with the compromise that some welfare might be lost.)
>
> > “What is the complexity for computing $\hat{\Theta_i}$ from $\tilde{\Theta_i}$?”
>
> This is an interesting question for future research that we have not explored in full generality in the present work. However, in the case that $\widetilde{\Theta}_i$ is a polytope (as in Theorem 3.6), then $\widehat{\Theta}_i$ obtained in the main random-expansion mechanism is also a polytope, and hence payments with respect to $\widehat{\Theta}_i$ can be computed via a similar LP to the one in the proof of Theorem 3.6. We will include a brief discussion of this point – thank you for bringing it to our attention!

---

> > ### Comment · Reviewer_kPNZ · 2023-08-18
> > **Thank you!**
> >
> > We would like to thank the authors for addressing our questions. We have no other questions at this point.

---

### Official Review · Reviewer_A5XS · 2023-07-24

**Soundness:** 3 good
**Presentation:** 4 excellent
**Contribution:** 3 good
**Rating:** 5
**Confidence:** 3

**Summary:**

This work focuses on studying social welfare and revenue maximization in multi-dimensional auctions under predictions. Each agent is associated with a private type $\theta_i$ drawn from a distribution $\Theta_i$​, and the mechanism designer utilizes predictions $\tilde{\theta}_i \in \Theta_i$​ as additional side information for the agent types. These predictions offer insights, such as constraining the sum of values for multi-item auctions to be within a constant limit. The authors' objective is to design incentive-compatible mechanisms that ensure both consistency, the worst-case multiplicative error when predictions are accurate, and robustness, the worst-case multiplicative error regardless of quality of the side informations.

The primary focus of the authors centers around maximizing social welfare and obtaining maximum revenue through efficient mechanisms. On the surface, the problem may appear straightforward, as a simple convex combination of trusting or discarding predictions achieves satisfactory consistency and robustness bounds. Specifically, a probability $\beta$ is used to decide whether to discard or fully trust the predictions, ensuring 1,$\beta$ consistency and robustness for social welfare and $(1−\beta),\beta$ consistency and robustness for revenue. However, the paper highlights that the mechanism's revenue significantly decreases when predictions are even slightly inaccurate. Therefore, the main objective of the paper is to generalize the robustness/consistency approach and analyze the rate of revenue degradation as predictions become invalid.

The principal contribution of the paper lies in its novel mechanism that goes beyond the binary decision of discarding or trusting predictions. Instead, it introduces a third option of randomly expanding the predictions. The mechanism starts from the predicted type space and considers all types $\theta \in \Theta_i$ within the $l_{\infty}$ ball around $\theta$ with a radius r, where r is randomly chosen from a discretization of the ambient type space's diameter. The paper's main result is achieved through an appropriate convex combination of these three options. Additionally, the authors explore a special case where agent types lie on the line or subspaces and derive corresponding guarantees.

**Strengths:**

The paper is well-written and effectively presents its contributions.
The introduction of the side information generalize the types of predictions in the standard consistency/robustness framework.

**Weaknesses:**

While the expansion of predictions is interesting, the results might not be particularly surprising.
Note that the work focusing on achieveing optimal revenue conditioning on maximizing welfare. This imposes significant restrictions on the revenue benchmark used for deriving theoretical guarantees.

**Questions:**

Is there any guarantees on revenue if the benchmark is the optimal revenue instead of the optimal revenue with respect to welfare-maximizing mechanisms?

---

> ### Author Rebuttal · Authors · 2023-08-09
>
> Thank you for your review! We respond to your questions and comments below.
>
> First, we would like to correct a major misunderstanding of our main results, highlighted by the following comment:
>
> > "Note that the work focusing on achieveing optimal revenue conditioning on maximizing welfare. This imposes significant restrictions on the revenue benchmark used for deriving theoretical guarantees.”
>
> We would like to clarify that our revenue benchmark is not conditioned on efficiency. Our mechanisms optimize revenue directly, and the revenue benchmark is the total social surplus (a.k.a. efficient/max welfare a.k.a. OPT in our paper). We do not require efficiency nor do we always get full efficiency. Total social surplus is the *strongest* possible benchmark for both welfare and revenue, and in particular optimal revenue is always bounded above by total social surplus.
>
> > “Is there any guarantees on revenue if the benchmark is the optimal revenue instead of the optimal revenue with respect to welfare-maximizing mechanisms?”
>
> As mentioned in the previous comment, our revenue benchmark is total social surplus, which is the strongest possible revenue benchmark (even stronger than optimal revenue).
>
> > “The mechanism starts from the predicted type space and considers all types within the ball around with a radius r…The paper's main result is achieved through an appropriate convex combination of these three options.”
>
> Our main mechanism on page 7 is the random expansion mechanism. It does not involve a convex combination of any other mechanisms. What makes the random expansion mechanism powerful is the ability to choose the parameters $\zeta = (\zeta_1,\ldots, \zeta_n)$ and $\lambda = (\lambda_1,\ldots,\lambda_n)$. If the side information is high quality and the mechanism designer chooses $\zeta$ and $\lambda$ wisely, both welfare and revenue of our mechanism approach the total social surplus OPT.
>
> Based on this comment we are led to believe that you might have read an old arXiv version of this paper instead of the present submission to NeurIPS-23. If this is indeed the case, we kindly ask that you update your score based on your review of the current submission.

---

### Decision · Program_Chairs · 2023-09-21

**Decision:**

Accept (poster)

**Comment:**

Overall, reviewers are positive about the submission. They think the paper studies an interesting problem in mechanism design and the paper is well-written.

The major concern reviewers have is that, despite the side information setting is motivated by practical applications, the paper lacks empirical studies and it is unclear how practically applicable the paper's result is.